# Abruptly attenuated carbon sequestration with Weddell Sea dense waters by 2100

Cara Nissen [1✉], Ralph Timmermann[1], Mario Hoppema [1], Özgür Gürses[1] & Judith Hauck [1]

Antarctic Bottom Water formation, such as in the Weddell Sea, is an efficient vector for carbon sequestration on time scales of centuries. Possible changes in carbon sequestration under changing environmental conditions are unquantified to date, mainly due to difficulties in simulating the relevant processes on high-latitude continental shelves. Here, we use a model setup including both ice-shelf cavities and oceanic carbon cycling and demonstrate that by 2100, deep-ocean carbon accumulation in the southern Weddell Sea is abruptly attenuated to only 40% of the 1990s rate in a high-emission scenario, while the rate in the 2050s and 2080s is still 2.5-fold and 4-fold higher, respectively, than in the 1990s. Assessing deep-ocean carbon budgets and water mass transformations, we attribute this decline to an increased presence of modified Warm Deep Water on the southern Weddell Sea continental shelf, a 16% reduction in sea-ice formation, and a 79% increase in ice-shelf basal melt. Altogether, these changes lower the density and volume of newly formed bottom waters and reduce the associated carbon transport to the abyss.

[1] Alfred Wegener Institut, Helmholtz Zentrum für Polar- und Meeresforschung, Bremerhaven, Germany. ✉email: cara.nissen@awi.de

Due to its unique setting in the global overturning circulation[1], the Southern Ocean plays an important role in the partitioning of carbon between the atmosphere and the deep ocean. Early box models[2–4] and later also general ocean circulation models[5] demonstrated that the formation of Antarctic Bottom Water (AABW) at southern high latitudes facilitates carbon sequestration on centennial to millennial time scales[6], which in turn exerts a strong control on global atmospheric $CO_2$ concentrations and climate. Despite this recognized importance on climatically relevant time scales[2–4], no quantitative information exists to date about the amount of carbon sequestered during the formation of AABW and its importance relative to deep carbon accumulation through sinking biotic particles[6]. The difficulty of ocean models to correctly simulate all processes involved in AABW formation[7,8] and the scarcity of observational data in its formation regions on the Antarctic continental shelves[9] complicate both the assessment of present-day AABW formation and carbon sequestration rates, as well as the detection of climate-change impacts. This is true even for the Weddell Sea in the Atlantic sector of the Southern Ocean, which since the 1930s has been recognized as the most important AABW formation region[10,11] and is the best-observed one to date[12].

Generally, the formation of AABW can be divided into two steps[13–16]: First, water flowing into the southern Weddell Sea on the eastern flank of the cyclonic Weddell Gyre[17,18] is transformed to Dense Shelf Water (DSW) as a result of the buoyancy loss caused by atmosphere–ocean, sea ice–ocean, and ice shelf–ocean interactions[14] (Fig. 1). Most of the density increase occurs on the southwestern Weddell Sea continental shelf as a result of heat loss and local sea-ice formation; it is to some extent counteracted by precipitation and meltwater fluxes from the Filchner-Ronne and Larsen ice shelves[19–24]. In particular, the densest variety of DSW, i.e., High Salinity Shelf Water, which is formed due to surface

cooling and brine rejection in sea-ice formation, is modified in the ice-shelf cavity to form the less saline, but colder Ice Shelf Water, which then mixes with the Low Salinity Shelf Water mostly prevailing on the southeastern Weddell Sea continental shelf[15,19]. Second, this resulting DSW cascades down the continental slope forming either Weddell Sea Deep Water (WSDW[17]; potential temperature referenced to 0 dbar $\theta_0$ between $-0.7\,°C$ and $0\,°C$) or Weddell Sea Bottom Water (WSBW[17]; $\theta_0 < -0.7\,°C$). Along the way, Warm Deep Water (WDW[17]; the Weddell Sea variant of Circumpolar Deep Water (CDW) with $\theta_0 > 0\,°C$) is entrained to ultimately form AABW, which then spreads throughout the abyss of the global ocean[13,14]. Integrating observations from 1973–2017, it has recently been suggested that, on average, $4.5 \pm 0.3$ Sv (Sverdrup; 1 Sv is $1 \cdot 10^6\,m^3\,s^{-1}$) of DSW formed on the Weddell Sea continental shelf in the first step entrain another $3.9 \pm 0.5$ Sv of WDW on its way to the abyss, resulting in a total of $8.4 \pm 0.7$ Sv of AABW transported northwards along the western flank of the Weddell Gyre[16].

AABW formation in the Weddell Sea is highly variable in time[25–27] and rather sensitive to changing environmental conditions due to its close ties with buoyancy fluxes and density distributions[14,27]. Both WSDW and WSBW have been warming and freshening since the late 1950s[28–34], which has at least partly been attributed to property changes of newly formed DSW[26,35], possibly due to changes in sea-ice formation and ice-shelf basal melt rates in the area[33]. For the 21st century, climate models project a slowdown or even a complete shutdown of AABW formation under the high-emission RCP8.5 scenario[7], as a consequence of continuous warming and freshening of high-latitude waters[36]. However, limited by the coarse resolution (~50 km) on the Antarctic continental shelves and the absence of an ice-shelf component, many of these models struggle to correctly reproduce AABW properties[8,37] and often form AABW entirely via spurious

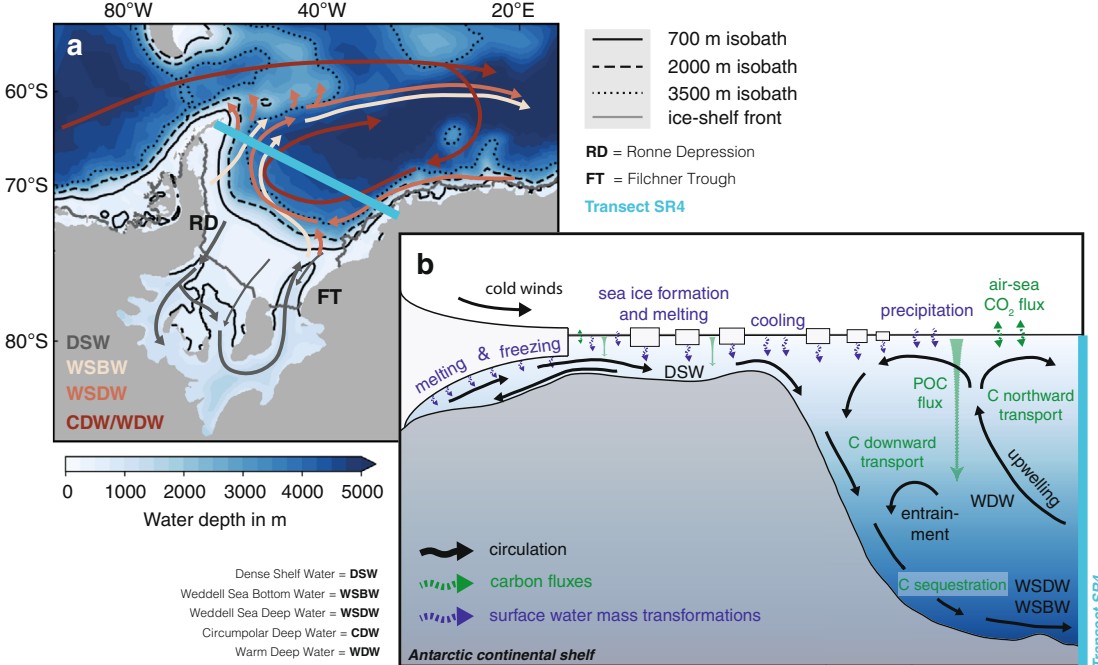

**Fig. 1 Sketch illustrating the major processes involved. a** Water depth in m below the ocean surface in the Weddell Sea and a schematic view of the general two-dimensional circulation in the area, adapted from ref. [9]. The transect SR4 of the World Ocean Circulation Experiment is marked in mint, and different water masses are distinguished by colors. **b** Typical section from the Antarctic continent to the transect SR4, with general features of the overturning circulation sketched in black, adapted from refs. [1,9]. Highlighted in blue are surface water mass transformations by buoyancy fluxes, and carbon fluxes are marked in green. CDW Circumpolar Deep Water, WDW Warm Deep Water, WSDW Weddell Sea Deep Water, WSBW Weddell Sea Bottom Water, DSW Dense Shelf Water, POC particulate organic carbon.

deep convection in the open ocean rather than via water mass transformations on the Weddell Sea continental shelves south of the World Ocean Circulation Experiment transect SR4, which connects the tip of the Antarctic Peninsula with the eastern Weddell Sea at Kapp Norvegia (Fig. 1)[7,8,38,39]. The reduction of AABW formation in response to an expected large freshwater discharge from ice shelves in the upcoming decades[40] has recently been confirmed with higher-resolution ocean models, which better capture AABW formation regions on the Antarctic shelf[41,42]. These model experiments were, however, highly idealized with prescribed freshwater input at the surface instead of the employment of a dedicated ice-shelf component in the model. While ocean models including a quantitative representation of ice shelf-ocean interaction project an up to 15-fold increase in Weddell Sea ice-shelf basal melt rates by the year 2100[43–48], the relative contribution of enhanced ice-shelf basal melt and reduced sea-ice formation[44,47] to changes in Weddell Sea DSW formation and the implications for carbon transfer to depth remain unquantified.

An improved understanding of Weddell Sea carbon cycling is urgently needed, as observations have revealed substantial changes in carbon cycling in this area over the recent past[9,49–53] : While the Weddell Sea was a net source of carbon to the atmosphere in preindustrial times[9,50], it is currently a net sink of ~33–80 Tg C per year[49,52,53]. Even though climate models suggest an up to 4-fold increase in $CO_2$ uptake in the high-latitude Southern Ocean by the end of the 21st century[54], it is still unclear how these projected changes in air-sea $CO_2$ exchange might affect carbon sequestration in the deep ocean.

Here, we fill this gap by using a global ocean–sea ice–biogeochemical model[55–59] with a representation of ice-shelf cavities[60] and an eddy-permitting resolution on the southern Weddell Sea continental shelves. Forcing our ocean model with atmospheric output from the AWI Climate Model[61], the comparison of a high-emission future scenario (simA) with a control simulation in a constant climate (simB) suggests a reduction in deep-ocean carbon transfer in the southern Weddell Sea towards the end of the 21st century. We show that this can mainly be attributed to changes in water mass properties and water mass transformations on the southern continental shelf, which result in a reduced connectivity between the upper and the deep ocean, thereby diminishing the carbon transfer to the abyss with newly formed dense waters.

## Results

**Deep-ocean carbon inventory and air-sea $CO_2$ exchange over the 21st century**. In the southern Weddell Sea south of the transect SR4, the total carbon inventory in the deep ocean below 2000 m increases by 0.75 Pg C between the years 1980 and 2100 under the high-emission SSP5-8.5 scenario (simA; dark grey line in Fig. 2a), which corresponds to 14% of the total increase in the whole water column in this area (compare to upper-ocean carbon inventory in Supplementary Fig. 1). However, the accumulation in the deep ocean is not steady throughout the 21st century. Acknowledging substantial decadal and interannual variability (Fig. 2c), the average deep-ocean accumulation rate of carbon amounts to 3.7 Tg C yr$^{-1}$ in the 1990s, increases to 9.4 Tg C yr$^{-1}$ and 14.7 Tg C yr$^{-1}$ in the 2050s and 2080s, respectively, and then abruptly declines to 1.5 Tg C yr$^{-1}$ in the 2090s (Fig. 2e). Over much of the 21st century, the deep-ocean carbon inventory also increases in the control simulation with a constant atmospheric $CO_2$ concentration and without climate-change forcing (simB; dashed grey line in Fig. 2b), demonstrating that the deep ocean of the southern Weddell Sea is not fully equilibrated with the constant atmospheric $CO_2$ concentration (313 ppm) in this

experiment, as expected from the chosen spin-up procedure (see Methods). Yet, the total increase by the year 2100 is small in simB (+0.27 Pg C) in comparison to simA (+0.75 Pg C), illustrating that the increase in simA is mostly attributable to the high $CO_2$ and climate-change forcing scenario. Further, as we will show in the following, the mechanisms causing the changes in the deep-ocean carbon inventory in simA are robust, as corresponding changes in simB are much smaller or even opposite in sign.

Even though the deep-ocean carbon accumulation rate in the 2090s in simA is lower than the average (±one standard deviation) over all decades (Fig. 2e), both decadal and interannual variability complicate the assessment whether the attenuation at the end of the 21st century is statistically significant (Fig. 2c). To resolve this, we apply the change point analysis, which uncovers any shifts in the mean or the linear trend of a given quantity that exceed natural variability (see Methods and refs. [62,63]). For the second half of the 21st century, this analysis reveals three change points in the linear trend in annual mean deep-ocean carbon accumulation rates (dark blue squares in Fig. 2c). While two change points mark the start (2073) and end (2084) of the high-accumulation phase in the late 2070s and the 2080s, the trend in the accumulation rate switches from positive to negative in the year 2050 (dark blue lines in Fig. 2c), as indicated by another change point at that time. This suggests that the reduced rates in the 2090s are the result of an underlying system change. Further, this implies that the high rates in the 2080s are a temporary phenomenon, interrupting the general trend towards lower accumulation rates. To what extent the trends or variability in deep-ocean carbon accumulation rates over the 21st century are controlled by changes in atmospheric $CO_2$ levels or other climate-related processes can be assessed by comparing simA to a simulation in which only atmospheric $CO_2$ levels vary, but not climate (simC; see Methods). In simC, deep-ocean carbon accumulation rates are overall much less variable than in simA, but are higher towards the end of the 21st century than in the 1990s (Fig. 2b, d, f). Notably, the accumulation rates increase after 2084 in simC, when the rates decline in simA. Altogether, this implies that climate-induced changes in carbon sequestration processes lead to most of the decadal and interannual variability in simA and explain the attenuated accumulation rates at the end of the 21st century, outweighing the increase in accumulation expected from increased atmospheric $CO_2$ levels alone. Still, this analysis leaves it unclear whether the overall reduction in simA after the year 2050 and the intermittent increase are caused by changes in the same mechanism or by the superposition of two mechanisms with different underlying trends or variability.

Any change in the deep-ocean accumulation rate of carbon can be caused by changes in the downward transfer of carbon via biological fluxes (i.e., sinking particles and carbon fluxes at the sediment–water column–interface), changes in the physical transport of carbon, or changes in the upper-ocean carbon inventory due to changes in air-sea $CO_2$ exchange. While the southern Weddell Sea remains a small source of $CO_2$ to the atmosphere until the year 2100 in the control simulation (~2 Tg C yr$^{-1}$, see Supplementary Fig. 1b), oceanic $CO_2$ uptake in this area in simA increases throughout the 21st century, amounting to 62 Tg C yr$^{-1}$ in 2100. As a consequence, also the upper ocean carbon inventory increases steadily (Supplementary Fig. 1a). At the same time, the deep-ocean carbon accumulation due to biological fluxes increases from 1.9 Tg C yr$^{-1}$ in the 1990s to 2.8 Tg C yr$^{-1}$ in the 2090s in simA (green bars in Fig. 2g). This increase is largely due to an enhanced sinking flux of particulate organic carbon with only a marginal contribution from sedimentary carbon release (Supplementary Fig. 2). The increased particulate organic carbon flux is the direct consequence of a 56% increase in biological productivity in the upper ocean south of

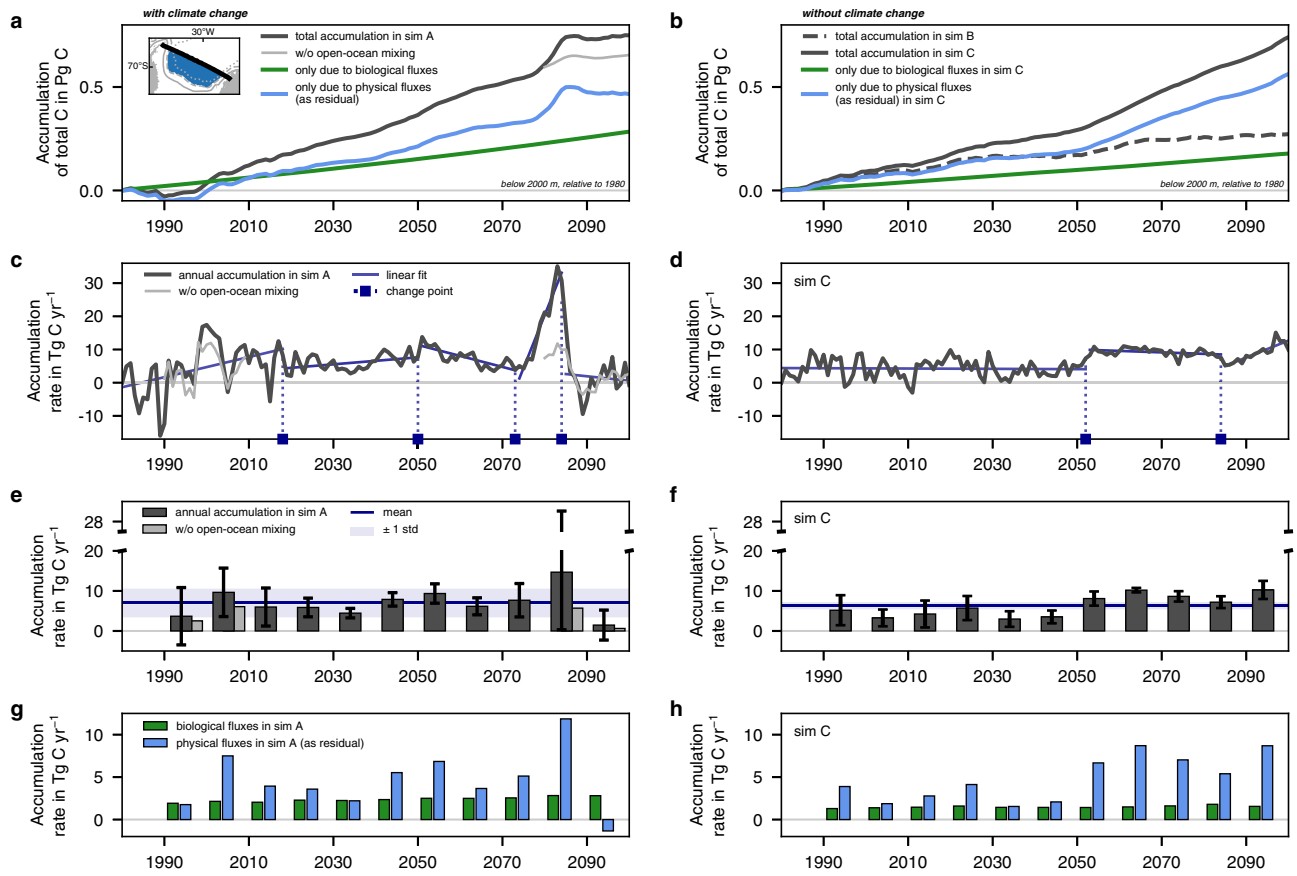

**Fig. 2 Deep-ocean carbon accumulation. a, b** Accumulated carbon in Pg C between 1980 and 2100 below 2000 m in the southern Weddell Sea south of the transect SR4 (see blue area in inlet) in **a** the model simulation *simA* (dark grey; historical + SSP5-8.5 scenario) and **b** the control simulation *simB* (dashed dark grey) and *simC* (solid dark grey; constant climate + varying atmospheric $CO_2$). **c–f** Show carbon accumulation rates in Tg C yr$^{-1}$ averaged **c, d** annually and **e, f** for each decade in **c, e** *simA* and **d, f** *simC*. In **c, d**, the dark blue lines indicate the statistical models providing the best fits to describe the time series, and change points are indicated with vertical lines and as squares on the x axis (see Methods and refs. [62,63]). In **e, f**, the whiskers depict one standard deviation within each decade, the horizontal blue line indicates the average accumulation rate between 1990–2100, and the shading shows ± one standard deviation around the mean. The light grey lines in **a, c** and the light grey bars in **e** denote the accumulated carbon due to all processes except vertical mixing in the open ocean north of the 3500 m isobath (dotted line in the map in **a**). The contribution of biological and physical fluxes (the latter calculated as residual) in **a, g** *simA* and **b, h** *simC* are shown in green and blue, respectively. Biological fluxes are dominated by sinking particle fluxes (Supplementary Fig. 2).

SR4, in response to warming and higher light availability due to the shrinking summer sea-ice cover by 2100 (Supplementary Fig. 1c–e). Altogether, the steady increases in oceanic $CO_2$ uptake, in the upper-ocean carbon inventory, and in the downward carbon transfer with the biological pump imply that these fluxes cannot explain the simulated decline in deep-ocean carbon accumulation in the 2090s. Instead, this decline is dominated by changes in the physical fluxes of carbon in the model. While the deep-ocean carbon accumulation due to physical transport increases from 1.8 Tg C yr$^{-1}$ in the 1990s to 11.9 Tg C yr$^{-1}$ in the 2080s, it dwindles and even abruptly changes sign in the 2090s, when all physical fluxes combined constitute a net loss of carbon from the deep ocean (blue bars in Fig. 2g). This evolution of physical fluxes in *simA* is in stark contrast to the one in *simC*, in which physical fluxes transfer an increasing amount of carbon to the deep ocean throughout the 21st century (Fig. 2h). Taken together, by the end of the 21st century, the deep-ocean carbon inventory in *simA* continues to increase—albeit at a much smaller rate compared to the previous decades—only due to the biological fluxes.

**Disentangling the physical flux components contributing to deep-ocean carbon accumulation.** In general, changes in the

deep-ocean carbon accumulation rate due to physical transport can be due to lateral or vertical fluxes via advection or mixing. For the deep-ocean carbon budget considered here, the divergence of vertical fluxes typically dominates over the contribution by lateral fluxes (Fig. 3), as gradients in total carbon concentrations in the vertical exceed those in the lateral (compare Fig. 3d to Supplementary Fig. 3). In our model, vertical mixing is the dominant flux component in the southern Weddell Sea (Fig. 3a), with all others being at least one order of magnitude smaller (e.g., vertical and lateral advection; Fig. 3b, c; see also Supplementary Figs. 4 and 5). Consequently, vertical mixing as a signature of convection accounts for >90% of the total physically-driven carbon accumulation throughout much of the 20th and 21st century, and this flux component alone can explain most of the simulated evolution in deep-ocean carbon accumulation (compare Fig. 3a to Fig 2g). While the slope region between the 2000 m and 3500 m isobaths, which constitutes the pathway of newly formed dense waters from the continental shelf to the deep ocean (especially in the southwest; Fig. 1), contributes up to 50% to the downward transfer of carbon in the 2000s, the enhanced downward transfer in the 2080s (~8 Tg C yr$^{-1}$) can largely be attributed to enhanced open-ocean mixing (Fig. 3a, light grey lines in Fig. 2a, c, and Supplementary Fig. 4), with only minor contributions from other

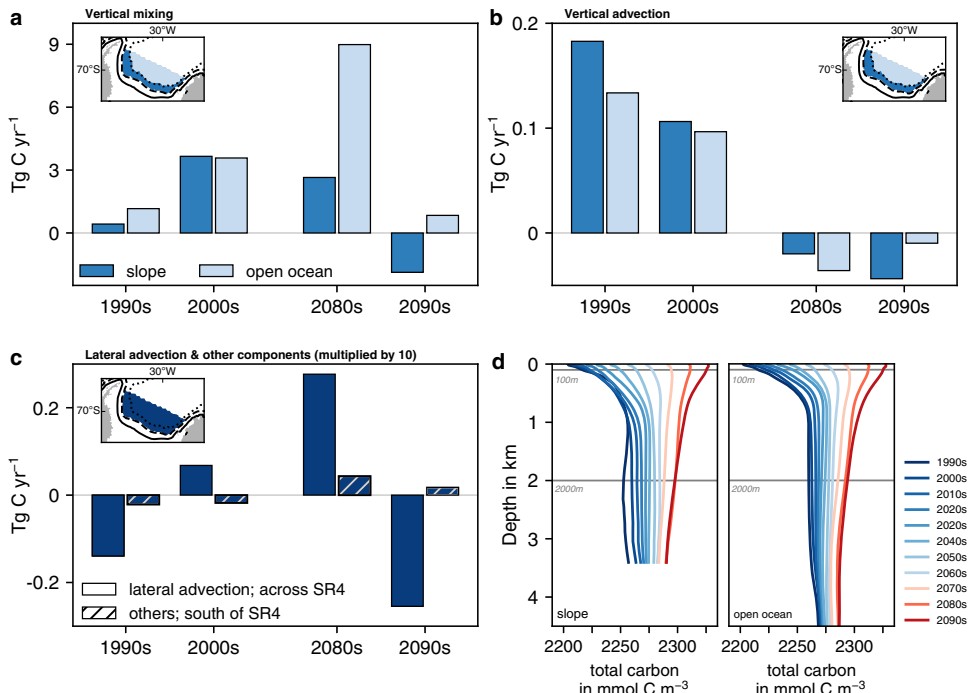

**Fig. 3 Divergence of physical flux components contributing to changes in the deep-ocean carbon inventory.** Changes in the total carbon inventory south of the transect SR4 and below 2000 m in the model simulation *simA* (historical + SSP5-8.5 scenario) in the 1990s, 2000s, 2080s, and 2090s due to **a** vertical mixing across 2000 m, **b** vertical advection across 2000 m, and **c** lateral advection across the transect SR4 and the sum of all other flux components, e.g., from the eddy parametrization (dashed bars). Positive fluxes denote an increase in the inventory in the volume of interest due to the respective flux component. All fluxes are in Tg C yr$^{-1}$, but note that the fluxes in **c** are one order of magnitude smaller than those in **a**, **b**. **d** Shows vertical profiles of total carbon concentrations in mmol m$^{-3}$ averaged over each decade from 1990–2100 in *simA*. Profiles are colored as a function of time in shades of blue and red depending on whether concentrations at 100 m are lower (blue) or higher (red) than those at 2000 m, respectively. Vertical fluxes in **a**, **b** and vertical profiles in **d** are shown for the continental slope and the open ocean, which are separated at the 3500 m isobath (see inlet in **a**, **b**).

components (Fig. 3b, c and Supplementary Fig. 5). In the 2090s, a much reduced open-ocean downward mixing flux of carbon ( < 1 Tg C yr$^{-1}$) is outweighed by an upward mixing flux along the slope and the two advective flux components, which all act to reduce the deep-ocean carbon inventory and thus explain the net loss of carbon from the deep ocean by physical fluxes in this time period (blue bars in Fig. 2g). The shift to an increasingly upwards-directed advective flux is particularly pronounced along the southwestern continental slopes (Supplementary Fig. 6) and is in line with enhanced upwelling of deep waters resulting from the intensified upper ocean gyre circulation at the end of the 21st century (see barotropic stream function in Supplementary Fig. 5 and refs. [9,64]).

In general, towards the late 21st century, an increasing amount of carbon is transferred to the deep ocean for any given open-ocean mixing event, as carbon concentrations in the upper ocean exceed those in the deep ocean after the 2060s (reddish colored vertical profiles in Fig. 3d). In fact, without open-ocean mixing, total deep-ocean carbon accumulation between 2080 and 2100 would be 59% lower (light grey lines in Fig 2a, c). The residual average carbon accumulation rate between 2080–2100 (3.1 Tg C yr$^{-1}$) can largely be explained by biological fluxes (90%), implying continuously low deep-ocean carbon transfer between 2080–2100 with Weddell Sea dense waters. Irrespective of that, open-ocean mixing events represent positive anomalies in deep-ocean carbon accumulation, temporarily overriding the underlying shift towards a reduced downward carbon transfer with dense waters along the southwestern continental slopes in the Weddell Sea (Fig. 2c). For example, the enhanced open-ocean downward flux of carbon in the 2080s is the result of the downward mixing of relatively recently ventilated and thus

carbon-enriched dense waters originating from regions upstream of the southern Weddell Sea (Supplementary Figs. 6 and 7 and Supplementary Section 2). Overall, the dominance of vertical physical fluxes in controlling the variability in carbon transfer to depth—in particular the attenuated transfer in the 2090s—implies that deep-ocean carbon accumulation rates are sensitive to physical processes in the overlying water column and upstream on the Weddell Sea continental shelves.

Indeed, bottom waters descending the continental slope in *simA* are less well connected to the upper ocean towards the end of the 21st century than they were before (Fig. 4a, c), which is explained in the following. To assess the connectivity between the deep and the upper ocean, we use an age tracer, which tracks any water parcel's last contact with the atmosphere–ocean, sea ice–ocean, or ice shelf-ocean interface as it mixes with surrounding waters in the ocean interior in our model simulations (see Methods). Everywhere on the continental shelf south of the 700 m isobath (solid black line in Fig. 4a), bottom waters are well ventilated in the 1990s, when >80% of the bottom waters have been in touch with one of these interfaces since the start of the simulation (purple colors in Fig. 4a). In comparison, bottom waters in areas deeper than 2000 m (dashed black line in Fig. 4a) are far less well ventilated in the same decade both on the continental slope (50%) and in the open ocean (<30%), i.e. north of the 3500 m isobath (green colors north of the dotted black line in Fig. 4a). This is further illustrated by the far better mixed water column on the Weddell Sea continental shelf compared to in the open ocean (Supplementary Fig. 8), implying longer time scales associated with bottom water ventilation in the open ocean than on the continental shelf. In the 2090s, the ventilation rate remains largely unchanged on the continental shelf, but declines

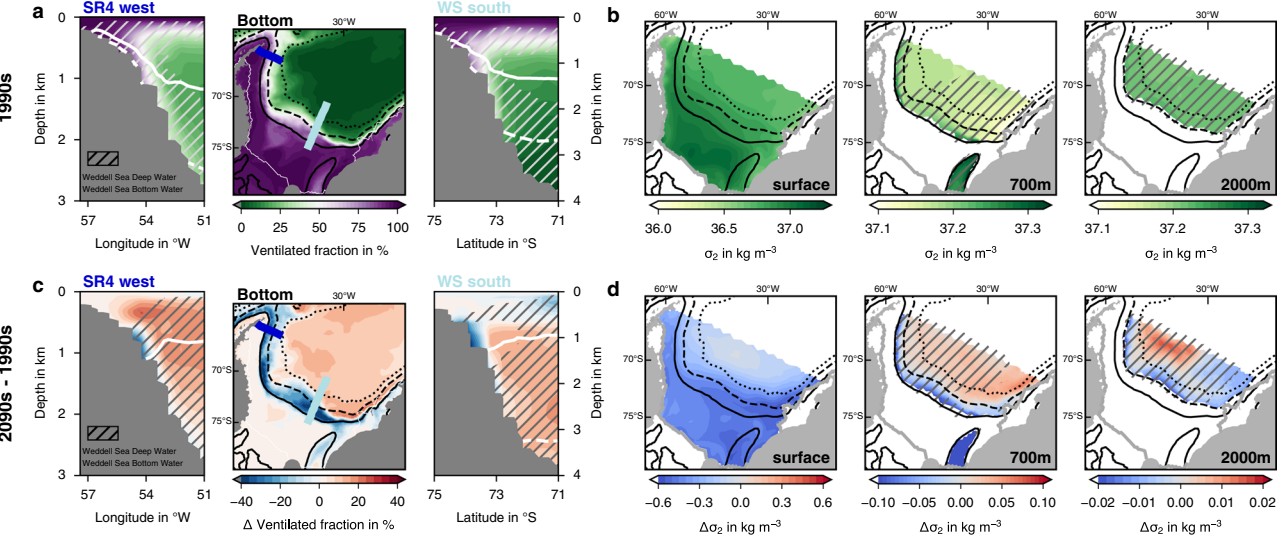

**Fig. 4 Bottom water ventilation and density distributions. a** Age-tracer-based fraction of water that is ventilated at the bottom (middle) and across the continental slope in the western (SR4 west; left) and southern Weddell Sea (WS south; right) in the 1990s of *simA* (historical + SSP5-8.5 scenario). The positions of the transects are indicated in the map as the dark blue (SR4 west) and the light blue line (WS south), respectively. **b** Distribution of potential density anomalies ($\sigma_2$ in kg m$^{-3}$, i.e., potential density referenced to 2000 dbar minus 1000 kg m$^{-3}$) at the surface, 700 m, and 2000 m in the 1990s. Note the different scales. **c, d** Same as **a, b**, but for the change in **c** the ventilated fraction and **d** $\sigma_2$ between the 2090s and the 1990s. In the transects in **a, c**, $\sigma_2$ isolines for the **a** 1990s and **c** 2090s are shown as the solid (37.2 kg m$^{-3}$) and dashed (37.25 kg m$^{-3}$) white contour, respectively. In all maps, black contours show the 700 m (solid), 2000 m (dashed), and 3500 m (dotted) isobaths. Hatching denotes the presence of Weddell Sea Deep Water and Weddell Sea Bottom Water in the **a, b** 1990s and **c, d** 2090s, defined in the model as waters with a potential temperature <−0.2 °C and a practical salinity >34.55.

profoundly along the slope (locally by more than 40%, Fig. 4c), demonstrating a reduced connectivity between the continental shelf and the deep ocean by the year 2100.

Changes in the density distribution support this finding: While the densities at 700 m and 2000 m in the open ocean mostly increase in response to the intensification of the gyre circulation and the corresponding increase in upwelling over the 21st century (Supplementary Fig. 5), densities close to the continental slope decline by up to 0.2 kg m$^{-3}$ and 0.033 kg m$^{-3}$ at 700 m and 2000 m, respectively (potential density referenced to 2000 dbar; Fig. 4d). In fact, while isopycnals of high density, i.e., $\sigma_2 > 37.2$ kg m$^{-3}$, are connected to the continental shelf sea in the 1990s (white isolines in Fig. 4a), providing a pathway of WSDW and WSBW into the abyss, this connectivity is absent by 2100 (Fig. 4c). As the atmospheric warming in the southern Weddell Sea accelerates towards the end of the 21st century (Fig. 5a), also the connectivity of bottom waters between the continental shelf and the open ocean decreases at a higher rate from the 2070s onwards (see Fig. 5c and change points in Fig. 5b). While density difference of bottom waters in the open ocean and those on the southern continental shelf amounts to 0.11 kg m$^{-3}$ in the 1990s and 0.23 kg m$^{-3}$ in the 2060s, it increases to 0.44 kg m$^{-3}$ by the year 2100 (dark blue line in Fig. 5b). As a consequence, this implies major changes in water mass properties and transformations on the southern Weddell Sea continental shelves, where new dense waters are typically formed.

**Changes in water mass properties and dense water formation on the Weddell Sea continental shelf.** Under the high-emission SSP5-8.5 scenario, water mass properties are projected to change throughout the water column on the southern Weddell Sea continental shelf (Fig. 6). While the decline in density is largest at the surface ($\Delta\sigma_2 < -0.5$ kg m$^{-3}$ at the continental shelf break along the 700 m isobath; Figs. 6f and 4d), the average decline in bottom density on the shelf is still significant, amounting to

0.28 kg m$^{-3}$ in the 2090s (Fig. 6a). This decline is due to a combination of a pronounced freshening (−0.31 in salinity; Fig. 6b) and warming (+0.35 °C; Fig. 6c) of bottom waters by the end of the 21st century. Concurrently, the total heat content on the shelf increases by 66% (Fig. 6d), reflecting both an increased presence of WDW on the shelf and particularly the reduction of heat loss of shelf waters to the atmosphere, as the warming of offshore WDW is not sufficient to explain the warming of continental shelf waters (Supplementary Fig. 9). Most of these simulated changes can be attributed to the climate-change forcing scenario, as trends in these properties in the control experiment *simB* are much smaller or even opposite in sign (light grey bars in Fig. 6a–d). As a result, keeping the definition of the water masses unchanged, less DSW has temperature (<−0.2 °C) and salinity (>34.55; see also Methods) properties of WSDW or WSBW in the 2090s as compared to in the 1990s. While most waters at the continental shelf break fulfill these criteria in the 1990s (61% of the total ocean area; Fig. 6e), the area covered by these dense waters has shrunk to only 13% by the year 2100 (Fig. 6f), thereby ultimately affecting the water mass properties of newly formed AABW in the southern Weddell Sea. In fact, the reduction in bottom density at the shelf break is most pronounced where shelf water is exported to the abyss, namely in the Filchner Trough and in the northwestern Weddell Sea (−0.3 kg m$^{-3}$; Fig. 6f), and is smallest in the Ronne Depression (−0.1 kg m$^{-3}$), where most newly formed dense waters enter the ice-shelf cavity and undergo freshening before leaving the shelf via the Filchner Trough (see Fig. 1 and ref. [19]). Therefore, the overall lower density of DSW at the end of the 21st century directly affects the transfer of these dense waters from the continental shelf to the deep ocean (see white isolines in Fig. 4a) and suggests significant changes in water mass transformation on the continental shelf.

The densification of waters in the southern Weddell Sea can be assessed in the water mass transformation framework, which relates surface density distributions to buoyancy fluxes at the atmosphere–ocean, sea ice–ocean, or ice shelf–ocean interface

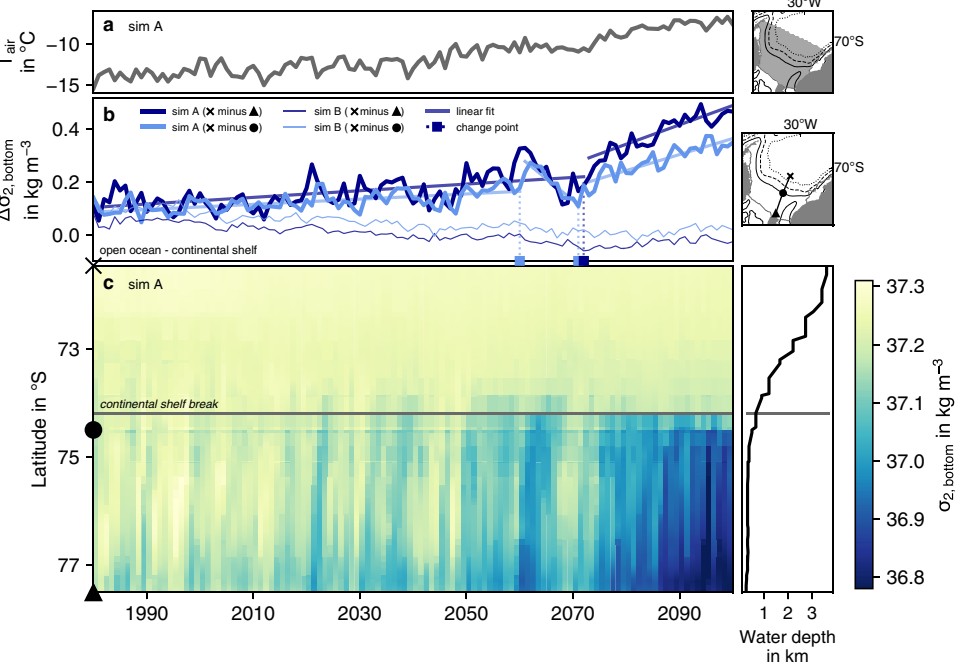

**Fig. 5 Evolution of bottom density between 1980 and 2100 in the southern Weddell Sea. a** Shows the average air temperature in °C over the southern Weddell Sea (see map) used to force *simA* (historical + SSP5-8.5). **b** Shows the difference in bottom potential density anomalies ($\sigma_2$ in kg m$^{-3}$, i.e., potential density referenced to 2000 dbar minus 1000 kg m$^{-3}$) between the open ocean and two locations on the continental shelf (light and dark blue, respectively) in *simA* (thick lines) and in the control simulation *simB* (thin lines). See symbols in the map for the exact locations used to compute the density differences. For *simA*, the straight lines indicate the statistical models providing the best fits to describe the time series, and change points are indicated with vertical lines and as squares on the x axis (see Methods and refs. [62,63]). **c** Indicates the corresponding evolution of bottom density along the whole transect in the southern Weddell Sea (see map) in *simA* (colors) and the bottom topography along this transect (black line). The location of the shelf break, defined as the 700 m isobath, is denoted by the horizontal grey line.

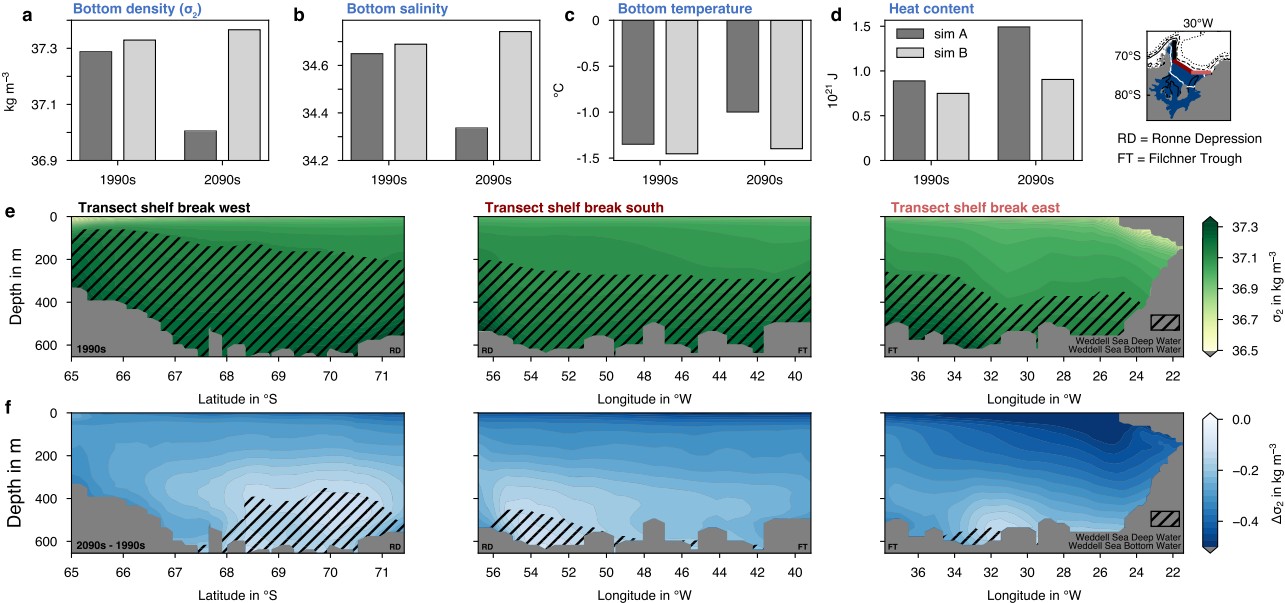

**Fig. 6 Changes in water mass properties on the Weddell Sea continental shelf. a** Bottom potential density anomaly ($\sigma_2$ in kg m$^{-3}$, i.e., potential density referenced to 2000 dbar minus 1000 kg m$^{-3}$) on the shelf (blue area in the map) in the 1990s and 2090s in the experiment *simA* (dark grey; historical + SSP5-8.5 scenario) and in the control simulation *simB* (light grey). **b–d** Same as **a**, but for **b** bottom salinity, **c** bottom potential temperature in °C, and **d** total heat content in 10$^{21}$ J. **e** Distribution of $\sigma_2$ at a transect along the continental shelf break (see map) in the 1990s of *simA*. **f** Same as **e**, but for the change in $\sigma_2$ between **f** the 2090s and the 1990s. Hatching in **e**, **f** denotes the presence of Weddell Sea Deep Water and Weddell Sea Bottom Water in the **e** 1990s and **f** 2090s, defined in the model as waters with a potential temperature <−0.2 °C and a practical salinity >34.55. The approximate positions of the Ronne Depression (RD) and the Filchner Trough (FT) are indicated.

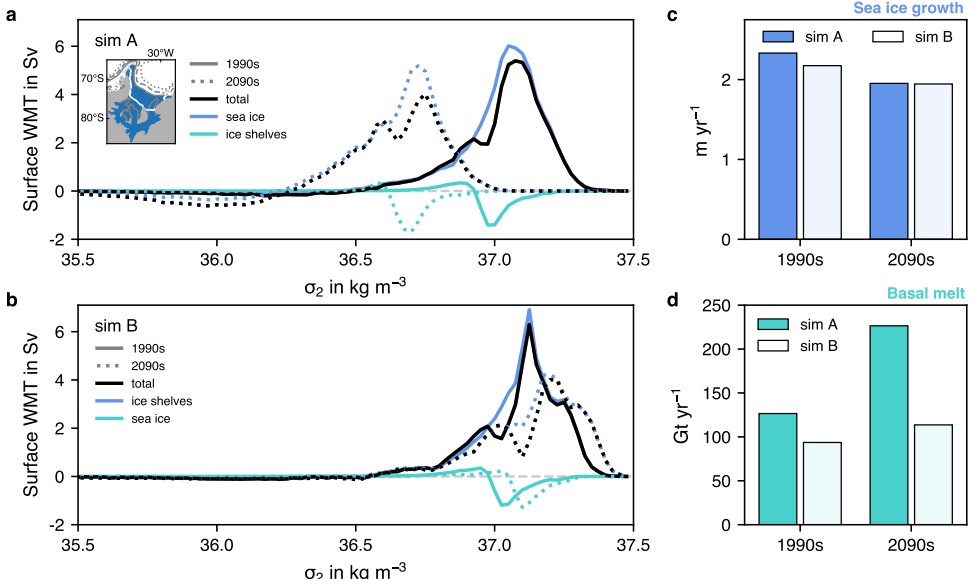

**Fig. 7 Water mass transformations due to surface buoyancy fluxes. a, b** Surface water mass transformation rates (WMT) in Sverdrup (Sv; 1 Sv is $1 \cdot 10^6$ m$^3$ s$^{-1}$) as a function of the potential density anomaly ($\sigma_2$ in kg m$^{-3}$, i.e., potential density referenced to 2000 dbar minus 1000 kg m$^{-3}$) due to the total buoyancy flux (black) and due to buoyancy fluxes from sea ice (blue) and ice shelves (mint) on the southern Weddell Sea continental shelf (see blue area in map) in the 1990s (solid) and 2090s (dotted) from **a** simA (historical + SSP5-8.5 scenario) and **b** the control simulation simB. Positive transformations denote a densification of surface waters due to buoyancy fluxes. Transformations due to heat fluxes and freshwater fluxes from evaporation minus precipitation are an order of magnitude smaller than those shown here (see Supplementary Fig. 10). **c** Sea-ice growth in m yr$^{-1}$ and **d** ice-shelf basal melt rates in Gt yr$^{-1}$ in the 1990s and 2090s in simA (darker colors) and simB (lighter colors).

and from which areas and formation rates of downwelling dense water masses can be derived (see Methods and refs. [42,65–68]). Total water mass transformation rates are largely positive on the Weddell Sea continental shelf (black lines in Fig. 7a), indicating an overall densification of surface waters by buoyancy fluxes in this area, which is in agreement with observations[17]. In the 1990s, the formation of dense waters on the southern Weddell Sea continental shelf amounts to 5.4 Sv in simA (peak in solid black line in Fig. 7a), fairly close to the recent observation-based estimate of $4.5 \pm 0.3$ Sv[16]. In this region, it can mostly be attributed to net sea-ice growth (blue line in Fig. 7a), with only small modifications by ice-shelf freshwater fluxes (mint line in Fig. 7a), which counter the densification of waters by sea-ice formation and without which dense-water formation in the area would be ~0.6 Sv higher. In response to the environmental change over the course of the 21st century, sea-ice formation is projected to be 16% lower and ice-shelf basal melting 79% higher in the 2090s than in the 1990s (Fig. 7c, d). Southern Weddell Sea dense water formation in the 2090s in simA is 1.4 Sv lower than in the preceding century (Fig. 7a), but this decline has to at least partially be attributed to model drift, as dense water formation in the control experiment simB shows a similar decline in magnitude in the area of interest (Fig. 7b). In contrast, the simulated shift of dense water formation to lighter density classes in simA towards the end of the 21st century can clearly be attributed to climate change (Fig. 7a, b). The reduced sea-ice formation and increased freshwater discharge from ice shelves in the southern Weddell Sea are the dominant reasons for this shift, but property changes in the source waters upstream also play a role. Waters on the eastern Weddell Sea continental shelf experience pronounced freshening and particularly warming by the year 2100 (−0.026 in salinity and +0.64 °C, respectively; see Supplementary Fig. 11), and the 40% lower sea-ice formation and 8-fold higher basal melt rates result in much lighter waters flowing into the southern Weddell Sea in the 2090s than in the 1990s (Supplementary Fig. 12). Thereby, these lighter source

waters amplify the shift towards lighter densities of newly formed dense waters in the southern Weddell Sea, where these become too light in our model experiments to sustain contemporary rates of carbon transfer to the deep ocean at the end of the 21st century.

## Discussion

Using a model setup that includes both ice-shelf cavities and a representation of the ocean carbon cycle, our results quantify the role of both physical and biological processes in deep-ocean carbon accumulation in the southern Weddell Sea. Over most of the 21st century, the carbon transfer to the abyss due to physical processes is up to four times higher than due to biological processes in the high-emission scenario, until the physically-driven transfer abruptly changes sign in the 2090s (Fig. 2). This is the result of an increasing decoupling of bottom waters on the continental shelves and the deep ocean since the mid-century (Figs. 2 and 5), with bottom water density declining most along pathways of dense water export from the southern continental shelves to the abyss (Filchner Trough; Figs. 1 and 6). Ultimately, Weddell Sea Deep and Bottom Waters are thereby effectively cut off from renewal with waters descending the continental slope, as the newly formed lighter dense waters end up at shallower depths.

Notably, the decoupling of bottom waters on the southern continental shelves and in the open ocean is sustained in two idealized 50-year-long model extensions, which were forced with constant near-end-of-century air temperatures (ext1 and ext2; see Supplementary Section 1, Supplementary Figs. 13–16, and Methods). While episodic open-ocean mixing events temporarily enhance deep-ocean carbon accumulation in our experiments, the scarcity of adequate observations prevents the evaluation of the present-day frequency of these events and their impact on the deep-ocean carbon budget of the southern Weddell Sea. Given their absence in simC (Fig. 2d), these open-ocean mixing events seem to mainly be triggered by climate variability. While it remains unresolved whether the frequency or duration of these

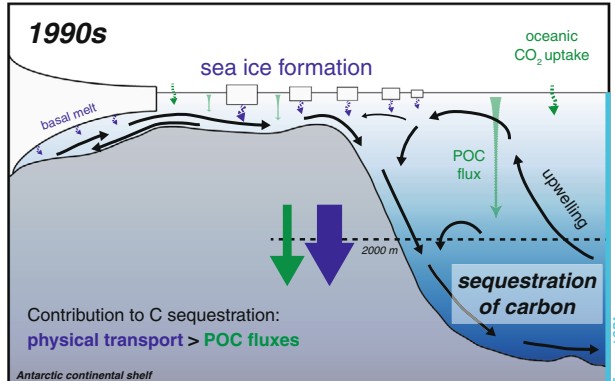
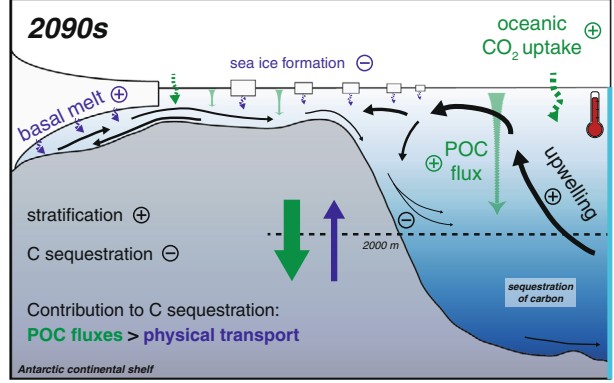

**Fig. 8 Sketch illustrating the simulated changes over the 21st century in the high-emission scenario.** Compared to the 1990s, deep-ocean carbon sequestration in the southern Weddell Sea is reduced by 60%, which is largely attributed to a reduction in the physical transport of carbon to the abyss. In particular, sea-ice formation is reduced by 16%, and ice-shelf basal melt increases by 79%, which both act to decrease the density of water on the continental shelf. Despite the 10-fold increase in oceanic $CO_2$ uptake and a 47% increase in sinking fluxes of biotic particles (POC), the decline in physically-driven downward transport dominates, as newly formed lighter dense waters on the continental shelf are transferred to shallower depths in the 2090s, reducing carbon accumulation in the deep ocean. In the sketch, font sizes and arrow thicknesses in the 2090s are scaled to approximately represent the magnitude of the simulated processes or fluxes.

events changes over the 21st century (due to the lack of physical flux output for some decades; see Methods), we note that the continued occurrence of these events in a warmer climate is in agreement with a recent study using another high-resolution model[39]. The view that the attenuation of carbon sequestration rates in the 2090s is abrupt implies a system change in the years directly leading up to this decade. Here, we interpret the fact that all physical fluxes combined act to transfer carbon upwards instead of into the deep ocean for the first time in this final decade of the 21st century as evidence for such a recent system change. In particular, compared to the physical fluxes throughout the 21st century, their change in the 2090s is drastic (Fig. 2g). As a result, the deep-ocean carbon accumulation in the 2090s is below the average rate over the whole analysis period ± 1 standard deviation for the first time in this final decade (Fig. 2e). Opposing to this view, the reduced sequestration rate in the 2090s can admittedly also be interpreted as a result of a gradual system change over a longer time period, with the high sequestration rates in the 2080s being a temporary anomaly (supported by the change point analysis; see Fig. 2). Nonetheless, whether gradual or abrupt, the system is clearly in a different state at the end of the 21st century.

Overall, the simulated decline in dense-water transfer to the deep ocean in response to enhanced stratification is in line with previous modelling experiments[7,39]. However, in contrast to some Earth System Models, which often exclusively form bottom waters via open-ocean mixing in areas north of the transect considered here[7], our model does not suggest a complete shutdown of dense water formation by 2100, but a less efficient transfer of newly formed dense waters from the Weddell Sea continental shelves to the abyss (Fig. 8). Although the downstream effects of the two mechanisms may be similar, namely less carbon sequestration in the high-latitude Southern Ocean, bottom water renewal is known to predominantly occur along the continental margins[13–16]. Thus, our model experiments display a more realistic representation of the mechanisms involved in dense water formation and transfer to the deep ocean, which is indispensable when aiming to anticipate their response to the ongoing environmental change.

In comparison to earlier modeling studies projecting an up to 15-fold increase in Weddell Sea ice-shelf basal melt rates[43–46], the projected 79% increase by 2100 in our model is much more moderate. In fact, our projected increase is much closer to the values recently reported by Naughten et al.[47,48], who—like in this

study—used atmospheric output from state-of-the-art climate projections to force their ocean model experiments. In their studies, ice-shelf basal melt rates in the Weddell Sea are projected to increase by 99% in the high-emission RCP8.5 scenario[47], and much higher atmospheric $CO_2$ levels than those considered in our study are required to induce an up to five-fold increase in basal melt[48]. Acknowledging that we have only used output from a single climate model here, our results are nevertheless in line with Naughten et al.[47,48], suggesting that earlier studies[43–46] using an older forcing scenario may have been too prone to WDW intrusions onto the continental shelf and hence overestimated the future basal melt rate increase in the area (see also discussion in ref. [48]). While a cold bias in WDW in our model (Supplementary Figs. 17 and 18) might imply that our experiments underestimate the response of basal melt rates to the climate-change forcing, the simulated present-day basal melt rates in the southern Weddell Sea (127 Gt yr$^{-1}$ in the 1990s) are within the range suggested by observations (192.6 ± 141.6 Gt yr$^{-1}$ for the ice shelves Larsen, Ronne, and Filchner[69]). In our model experiment, the increase in basal melt rates accelerates towards the end of the 21st century (143 Gt yr$^{-1}$ in the 2070s, 168 Gt yr$^{-1}$ in the 2080s, 226 Gt yr$^{-1}$ in the 2090s). Since the basal melt rate increases further in our model extension even under constant end-of-century temperatures (up to 288 Gt yr$^{-1}$ after 50 years in *ext1*; see Supplementary Fig. 16), basal melt rates can be expected to rise fast beyond 2100 under continued atmospheric warming, further freshening the shelf and subsequently reducing the density of newly formed dense waters and resulting in a continued low transfer of carbon to the abyss of the southern Weddell Sea.

As a result of the simulated changes in the physical environment (e.g., sea-ice cover), the transfer of carbon to the abyss with sinking biotic particles increases over the 21st century (Fig. 8). While the present-day biological pump adjacent to the southern Weddell Sea continental shelves is thought to be efficient in transferring biotic particles to the deep ocean[6,70], several studies have suggested rather shallow remineralization of those particles in the central Weddell Sea[9,53,71]. In particular, MacGilchrist et al.[53] have recently shown that biological fluxes in the Weddell Sea significantly contribute to carbon accumulation above 2000 m (CDW layer), but to a lesser extent to water masses below that, which is in line with our findings for the present time. Yet, by the year 2100, the carbon inventory of the southern Weddell Sea below 2000 m changes from being physically-controlled to being

biologically-controlled in our model experiment, which also holds true for the accumulation below 2500 m and 3000 m, respectively (Supplementary Fig. 19). This regime shift is dominated by the decrease in the physically-driven downward carbon transfer, which outweighs the increase in the biologically-driven transfer (due to more primary productivity and sinking particle fluxes) and which results in deep-ocean carbon accumulation rates in the 2090s amounting to only 40% of those simulated in the 1990s. However, we acknowledge that we might underestimate the future role of biotic particle fluxes in our model experiment. At high latitudes, phytoplankton growth is generally limited by the availability of micronutrients, such as iron, and the rather short growing season at these latitudes[9,72]. While the latter will likely be prolonged by 2100 as a result of the shrinking sea ice cover (Supplementary Fig. 1), iron supplied by meltwater from sea ice or icebergs was shown to be a source of iron for high-latitude phytoplankton[73], suggesting substantial fertilization as melt rates increase over the 21st century. This nutrient supply mechanism is currently not accounted for in our model, but might augment the biological response to environmental change.

Rates of carbon sequestration—if given relative to the $CO_2$ uptake at the ocean surface—are an indicator of the sequestration efficiency, but the sequestration depth is decisive for the degree of storage permanence, which describes how long the sequestered carbon will ultimately be locked away from the atmosphere. In the Weddell Sea, a high storage permanence with storage times on the order of centuries or more is likely if the carbon is transferred to the bottom of the ocean with WSBW along the continental slope[6,53]. In contrast, the storage time is likely much shorter if the carbon is only transferred to mid-depths, from where upwelling might bring the freshly sequestered carbon to the surface again within a few years or decades (especially in the central Weddell Sea where the open-ocean mixing events occur in our model experiment; see Fig. 3 and refs. [6,9,53]). Under the high-emission scenario considered in this study, the fraction of the oceanic $CO_2$ uptake sequestered below 2000 m in the southern Weddell Sea declines from 93% in the 1990s to only 2% in the 2090s (compare Fig. 2e to Supplementary Fig. 1b). This reduced sequestration efficiency implies that an increasing amount of carbon is stored at mid-depths and in the upper ocean over the 21st century (Supplementary Fig. 1). Indeed, most of the $CO_2$ taken up at the surface in the southern Weddell Sea throughout the 21st century evades long-term sequestration in the deep ocean, with total deep-ocean carbon accumulation until 2100 amounting to 26% of the total $CO_2$ uptake south of the transect SR4. While our two model extensions under constant end-of-century atmospheric forcing demonstrate that the evolution of carbon sequestration beyond 2100 is sensitive to the evolution of air temperatures, the decoupling between the continental shelf sea and the deep ocean is sustained for another 50 years even under the lower air temperatures in *ext2* (Supplementary Section 1). This demonstrates that more than a few decades and possibly also a larger reversal of the end-of-century warming than applied in *ext2* are needed to re-establish the pathway for dense waters from the continental shelves to the abyss in the southern Weddell Sea, which is in line with paleoceanographic literature[74–78].

On centennial and millennial time scales, a reduction in deep-ocean carbon accumulation will induce climate feedbacks as a consequence of the reduced storage time-scale. While these likely matter only marginally on the time scales considered in this study, justifying the use of an ocean-only model setup, longer simulations with fully coupled climate models including ice-shelf cavities should quantify both the magnitude of and the time scales associated with these feedbacks in future work. The AWI Climate Model used to force the experiments here projects an increase in Weddell Sea air temperatures under the SSP5-8.5 scenario

(4.1 °C) comparable to a multi-climate model mean (3.9 °C; 2015–2100; Supplementary Fig. 20). Given the large model spread (1.3–6.6 °C), using the atmospheric output of a different climate model would likely affect both the exact timing and the magnitude of the changes, but the impact on the controlling mechanisms identified here is expected to be small. Given that differences between ensemble members of the same model are typically smaller than differences between models, using different initial conditions in our simulations would likely only affect the exact timing of the simulated changes. In addition, the projected evolution of atmospheric variables depends on the emission scenario used. For the AWI Climate Model, the evolution of Weddell Sea air temperatures under the SSP5-8.5 scenario diverges from that under the SSP3-7.0 scenario only after the late 2070s (Supplementary Fig. 20). While the decoupling between the continental shelf sea and the deep ocean in *simA* in fact accelerates around that time (Fig. 5), the earlier onset of the decoupling in the mid-21st-century is in line with the onset of declining deep-ocean carbon sequestration rates (Figs. 2 and 5). This implies that a low-emission scenario might be needed to fully avoid the decoupling and the associated attenuation in deep-ocean carbon transfer with Weddell Sea dense waters. As this conclusion is possibly model-dependent, we note that due to large uncertainties associated with carbon-cycle feedbacks in climate models, the climate resulting from a higher-emission scenario in one model is also possible under lower-emission scenarios in another[79] (see also Methods).

In summary, the evolution of deep-ocean carbon sequestration in the southern Weddell Sea is closely tied to the evolution of buoyancy fluxes from sea ice and ice shelves and hence dense water formation on the continental shelves. Therefore, based on our results, the crossing of any critical threshold in high-latitude freshwater discharge, i.e., a tipping point above which newly formed dense waters are so light that they cease to reach the abyss[80], will likely be associated with a concurrent cessation in carbon sequestration. Additional research will be needed to elucidate whether such a tipping point has already been crossed by the end of our model experiment, or whether this abrupt decline is a non-permanent early warning signal of the imminent crossing of such a tipping point. Furthermore, it remains an open question whether such a tipping point can be avoided under lower-emission scenarios. In conclusion, the projected freshening of Weddell Sea shelf waters and the associated reduction in bottom water renewal reduce their capacity to store carbon on centennial and millennial time scales, with consequences for the partitioning of carbon between the ocean and atmosphere and hence the future evolution of climate.

## Methods

**Description of FESOM-REcoM**. We use the global Finite Element Sea Ice Ocean Model (FESOM) version 1.4[56] which includes a dynamic-thermodynamic sea-ice model[57] and an ice-shelf component[60]. Ice-shelf geometry and bottom topography are derived from RTopo-2[81] and prescribed to be constant in our model experiments. Coupled to FESOM is the Regulated Ecosystem Model version 2 (REcoM2)[55,58,59], which resolves the biogeochemical cycling of carbon, nitrogen, silicon, iron, and oxygen with a total of 28 prognostic tracers. The lower trophic level ecosystem is composed of two phytoplankton groups (silicifying diatoms and a mixed nanophytoplankton group, of which a fraction calcifies) and two zooplankton groups. Stoichiometric ratios are allowed to vary in REcoM2. Organic matter cycling is parametrized with one class of non-sinking dissolved organic matter and two size classes of particulate organic matter, which are remineralized as they sink through the water column. A fraction of the sinking particles reaches the sediment layer in the model, from which they are released back into the water column in dissolved form with fixed remineralization rates[55]. Consequently, any change in the deep-ocean carbon inventory due to biological fluxes is caused by changes in either the sinking particle flux across the depth horizon of interest (2000 m in this study) or in the release of dissolved inorganic carbon from the sediments, with the former dominating in the southern Weddell Sea (Supplementary Fig. 2). The *total carbon inventory* as assessed in this study refers to the sum of the following model tracers: dissolved inorganic carbon, dissolved organic carbon, two classes of sinking particulate organic carbon, particulate inorganic calcite, living

particulate organic carbon associated with the two phytoplankton and two zoo-plankton groups, and calcite associated with the nanophytoplankton group.

**Model setup, model assessment, and data sets for model evaluation.** For this study, all model experiments are run on a mesh with enhanced horizontal grid resolution on the continental shelves of the Southern Ocean, especially in the Weddell Sea. The horizontal resolution ranges from 4 km in the southern Weddell Sea to ~80 km at the outer edge of the Weddell Gyre and increases to >100 km outside of the Southern Ocean (Supplementary Fig. 21). The vertical is divided into 99 unevenly spaced z levels, of which 21 depth levels are situated below 2000 m. The time step in our simulations is 4 min. We run transient simulations with FESOM-REcoM for the years 1950–2100. At the ocean surface, we force the model with output from the AWI Climate Model (AWI-CM) produced for the "Coupled Model Intercomparison Project Phase 6 (CMIP6)"[61]. We use 3-hourly output of atmospheric momentum, radiation, and freshwater fluxes and daily output of terrestrial freshwater runoff from the first ensemble member of the historical simulation until 2014 and from the SSP5-8.5 scenario simulation thereafter. We note that the SSP5-8.5 scenario assumes a continuously high usage of fossil fuel for the evolution of $CO_2$ emissions[82], thereby assuming weaker climate protection policies for the 21st century than currently in place[79]. Nonetheless, the atmospheric $CO_2$ concentrations and the resulting future climate suggested by Earth System Models under the SSP5-8.5 scenario are possible even under lower-emission scenarios as a result of the large uncertainty associated with carbon-cycle feedbacks in these models[79]. Atmospheric $pCO_2$ levels are taken from Meinshausen et al.[83] for the historical period 1950-2014 and from O'Neill et al.[82] for the period 2015-2100, consistent with the time series used in the CMIP6 experiments. In our experiments, the physical tracers are initialized with FESOM output from the historical simulation of the AWI-CM and the biogeochemical tracers with REcoM output from an existing simulation for the "Regional Carbon Cycle Assessment and Processes 2 (RECCAP2)" project (unpublished). Thereby, at the start of our simulations, the physical and biogeochemical tracers have been spun up for 850 years (FESOM) and 100 years (REcoM), respectively.

For this study, three model experiments were performed: For the years 1950–2100, *simA* is forced with varying climate and varying atmospheric $pCO_2$ levels as described above, whereas *simB* is forced by repeating the atmospheric conditions of the year 1950 (atmospheric $pCO_2$; 312.82 ppm[83]) and 1955 (all other variables), allowing for an assessment of the model drift. The year 1955 was chosen aiming to (1) minimize the warming signal in the chosen year and (2) represent "normal" atmospheric conditions, which we identify by assessing the phase of the Southern Annular Mode (SAM) and El Niño Southern Oscillation in the first 20 years of the forcing. We computed seasonal anomalies of (a) the zonally averaged sea level pressure difference between 40°S and 65°S (SAM index) and (b) sea surface temperature between 5°N-5°S and 170°W-120°W in *simA* (Oceanic Niño Index; ONI). Ranking the years 1950–1969 according to a score, which comprises the annual mean SAM index, the annual mean ONI index, and the variability of these indices in each calendar year (expressed by one standard deviation across all seasons), resulted in the year 1955 being closest to "normal" conditions. We acknowledge that in the control simulation *simB*, the deep-ocean carbon inventory in the southern Weddell Sea is not in a steady state at the start of the analysis period in 1980 (Fig. 2b), which can be attributed to the spin-up procedure. In particular, atmospheric $CO_2$ concentrations have risen from the preindustrial 278 ppm to 313 ppm during the spin-up, and another ~100 years are needed in *simB* for the deep-ocean carbon inventory to reach quasi-equilibrium with this atmospheric $CO_2$ level (Fig. 2b). However, this does not affect the conclusions drawn in this study because upper ocean processes are close to equilibrium already in 1980 (see Supplementary Fig. 1) and because the climate-change signal in *simA* exceeds the drift in *simB* by far also for the deep ocean (Fig. 2a, b). Lastly, *simC* is forced with varying atmospheric $pCO_2$ levels (as in *simA*), but constant climate (as in *simB*). With this simulation, changes in deep-ocean carbon accumulation rates resulting from rising atmospheric $pCO_2$ alone, i.e., without the concurrent impact of a changing climate, can be quantified.

Further, given that carbon sequestration rates are attenuated towards the end of the 21st century in *simA*, this model experiment was extended by another 50 years in order to determine whether the reduced rates in the 2090s (a) represent a lasting change in the downward transfer of carbon with Weddell Sea dense waters or (b) can be explained by decadal variability. However, as atmospheric output from the AWI-CM is not available beyond the year 2100, the extension is forced with further increasing atmospheric $pCO_2$ levels according to the SSP5-8.5 scenario[82], but with constant climate forcing. We run two 50-year-long extensions which are forced by repeating the years 2091 (*ext1*) and 2095 (*ext2*), respectively. These 2 years were again chosen based on the state of the Southern Annular Mode and El Niño Southern Oscillation (see methodology for *simB*), in this case aiming to represent "normal" atmospheric conditions in the 2090s. Due to this experimental design, the simulated evolution of carbon sequestration rates for the 50 years in *ext1* and *ext2* constitute idealized cases for the considered high-emission scenario, as atmospheric warming would continue after the year 2100 rather than a temperature stabilization as assumed here under continuously increasing atmospheric $pCO_2$ levels. While being >5 °C higher than in the 1990s (−13.8 °C), the annual mean air temperature in the southern Weddell Sea in *ext2* (−8.4 °C) is lower than the one in *ext1* (−7.6 °C) and also than the decadal mean temperature for both the 2090s (−7.4 °C)

and the 2080s (−8.1 °C). This demonstrates the considerable interannual variability in air temperatures also in the 2090s, which superimposes the overall trend towards increasing air temperatures (Fig. 5a). As a result, Weddell Sea surface waters experience an overall lower air temperature in *ext2* than towards the end of *simA*, explaining the more frequent occurrence of vertical instabilities, which lead to enhanced deep-ocean carbon accumulation via open-ocean mixing (see Supplementary Section 1).

Model output is generally stored at monthly frequency, with physical flux output being only available for selected decades, namely between 1980–1999, 2080–2100, and for each decade of the model extensions. For the budget analysis of the deep-ocean carbon inventory in the southern Weddell Sea, vertical mixing fluxes are inferred from the change in the total carbon inventory, biological fluxes (see above), and the remaining physical flux components (lateral and vertical advection, horizontal diffusion, fluxes from the Gent-McWilliams parameterization, and the stabilization of the Taylor-Galerkin advection scheme[56]). For model evaluation, we use available ship-observations of dissolved inorganic carbon[84] and temperature & salinity[12] (see Supplementary Figs. 17, 18 and 22). Weddell Sea Deep Water (WSDW) and Weddell Sea Bottom Water (WSBW) in FESOM-REcoM are defined as waters with a potential temperature <−0.2 °C and a practical salinity >34.55, and we do not distinguish these two water masses in this study. The thresholds used here differ slightly from those in Fahrbach et al.[17], who suggested 0°C as the temperature threshold and 34.6 and 34.63 as the salinity thresholds to distinguish WSDW and WSBW, respectively, from Warm Deep Water (WDW). The definition was chosen here to best reflect the simulated water mass structure both south of and at the transect SR4, thereby accounting for a slight fresh bias in the focus region and a cold bias in the WDW core in the model (Supplementary Figs. 17 and 18).

**Age tracer.** Our model experiments include an age tracer, which is initialized at zero everywhere in the model domain. Over the course of the simulations, the age of a given water parcel increases accordingly and is only reset when it comes in contact with the surface, with *surface* here being either the air–ocean, sea ice–ocean, or ice shelf–ocean interface. The age tracer is advected and mixed as any other model tracer, implying that the age of a water parcel can also decline (increase) by mixing with a water parcel of lower (higher) age, i.e., if the latter water parcel has been in touch with the surface more recently (longer ago) than the former. Hence, based on a water parcel's age $a$, we calculate the ventilated fraction $\epsilon$ [%] of this water parcel at time $t$ of the simulation as

$$\epsilon(t) = 100 \cdot \frac{t - a_t}{t}. \tag{1}$$

With this definition, a given water parcel with an age corresponding to the simulation time $t$ is not ventilated at all, whereas an age of zero corresponds to 100% ventilated water. Consequently, a comparison of the age-tracer based ventilated fractions of bottom waters in the southern Weddell Sea between the 2090s and 1990s reveals changes in the exchange of these waters with the ocean surface.

**Water mass transformation framework.** Here, we are mostly interested in the formation of dense waters on the Weddell Sea continental shelves. To that aim, we assess water mass transformations due to surface buoyancy fluxes on the continental shelf south of the transect SR4 of the World Ocean Circulation Experiment, which connects the tip of the Antarctic Peninsula with the eastern Weddell Sea (Fig. 1), and in the eastern Weddell Sea (see Fig. 7 and Supplementary Fig. 12 for the exact location of both shelf regions). Any change in the volume of waters in a given density class between the surface outcrop of these waters south of the transect SR4 and their volume flux across the transect SR4 is due to either (1) a change in the transport of this density class across SR4, (2) diapycnal mixing fluxes in the ocean interior, or (3) surface transformations causing a lateral diapycnal flux[85]. In this context, the water mass transformation framework relates the density distribution at the ocean surface to buoyancy fluxes, providing a framework to relate the formation of dense waters to heat fluxes or freshwater fluxes from evaporation minus precipitation, sea ice, or ice shelves[42,65–68].

Discretizing the surface density ($\rho$) field into bins of 0.025 kg m$^{-3}$, heat fluxes $Q_{net}$ and freshwater fluxes $F_{net}$ transform water masses in the density bin $\rho_k$ with a rate $\Omega$ [Sv] at any time $t$ at the atmosphere–ocean or ice–ocean interface, following

$$\Omega(\rho_k, t) = -\frac{1}{\rho_{k+1} - \rho_k} \iint_A \frac{\alpha Q_{net}}{\rho_0 C_p} dA + \frac{1}{\rho_{k+1} - \rho_k} \iint_A \frac{\beta S F_{net}}{\rho_0} dA. \tag{2}$$

Here, $A$ is the outcrop area between the density bins $k$ and $k + 1$, $\alpha$ and $C_p$ are the thermal expansion coefficient and the heat capacity, respectively, $\beta$ and $S$ are the haline contraction coefficient and the surface salinity, respectively, $\rho_0$ is the reference density, and $F_{net}$ is composed of freshwater fluxes between atmosphere and ocean ($F_{Atm \to Ocean}$), sea ice and ocean ($F_{Seaice \to Ocean}$), and ice shelves and ocean ($F_{Iceshelf \to Ocean}$):

$$F_{net} = F_{Atm \to Ocean} + F_{Seaice \to Ocean} + F_{Iceshelf \to Ocean}. \tag{3}$$

The convergence (divergence) of waters in a given density class implies downwelling (upwelling) of these to satisfy mass continuity. We use the potential density referenced to 2000 dbar ($\rho_2$) as the density coordinate and display the density anomaly throughout this manuscript ($\sigma_2 = \rho_2 - 1000$ kg m$^{-3}$). While

different density coordinates have been used by other authors in the past[42,67], the qualitative results presented here are insensitive to this choice. Here, we compute water mass transformation rates in the southern Weddell Sea based on monthly model output for the 1990s and 2090s.

**Change point analysis**. In order to statistically identify (abrupt) changes in the simulated evolution of physical or biogeochemical quantities of the southern Weddell Sea over the 21st century, we use the R package "Detection of Structural Changes in Climate and Environment Time Series" ("EnvCPT", see refs. [62,63]). With this package, a number of pre-defined linear models is fit to a given time series, and the time series is screened for the existence of change points. Here, following the ref. [62], we include 8 statistical models in our analysis: a constant mean, a constant linear trend, a constant mean with first-order autocorrelation, a constant linear trend with first-order autocorrelation, multiple change points in the mean, multiple change points in the linear trend, multiple change points in the mean with first-order autocorrelation, and multiple change points in the linear trend with first-order autocorrelation. Subsequently, all model fits are ranked with the Akaike Information Criterion (AIC), which combines the maximum likelihood estimate for each statistical model, i.e., the optimal parameter set to describe the time series at hand, with a penalty for the number of parameters fitted[86]. Thereby, the statistical model with the lowest AIC score corresponds to the model that is best suited to describe the time series in question. We apply this methodology to the time series between 1980–2100 of annual mean deep-ocean carbon accumulation rates in simA and simC (Fig. 2c, d), air-sea $CO_2$ exchange in simA (Supplementary Fig. 1), and the difference in bottom density between the continental shelf and the open ocean in the southern Weddell Sea in simA (Fig. 5). For each of these quantities, the model including multiple change points in the linear trend is the best to describe the time series (with first-order autocorrelation for the dark blue line in Fig. 5 and without for all others). As we are mostly interested in decadal changes of annual mean properties in this study, we use a minimum segment length of 10 in the analysis, i.e., there has to be at least 10 years in between any two change points identified. We note, however, that the exact outcome of the change point analysis, including the timing of any change points, might be sensitive to this choice[62]. In particular, while the identified change points in the linear trend of deep-ocean carbon accumulation rates in simA (see Fig. 2c) are unaffected by choosing a minimum segment length <10, they change if larger intervals are chosen (e.g., change points in the mean with first-order autocorrelation are identified in 2020 and 2076 for a segment length of 12 or 15). This sensitivity can be understood by the length of the high-accumulation phase in the late 2070s and the 2080s of approximately 10 years, which prevents the identification of the onset and end of this event as individual change points if the required minimum segment length is > 10.

## Data availability

The minimal data set required to reproduce the findings of this study has been deposited on PANGAEA (https://doi.org/10.1594/PANGAEA.943505; see ref. [87]). Full model output is available upon request to the corresponding author.

## Code availability

The Fortran source code of FESOM1.4-REcoM2 can be obtained via https://fesom.de/models/fesom14/(last access April, 25, 2022). The analysis of model output was done with the open-source software Python.

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

## Acknowledgements

C.N. and M.H. were supported by the European Union's Horizon 2020 research and innovation programme under grant agreement No 820989 (project COMFORT), J.H. and O.G. were supported by the Initiative and Networking Fund of the Helmholtz Association (Helmholtz Young Investigator Group Marine Carbon and Ecosystem Feedbacks in the Earth System [MarESys], grant number VH-NG-1301), and R.T. was supported by the Helmholtz Climate Initiative REKLIM (Regional Climate Change), a joint research project of the Helmholtz Association of German research centres (HGF). The work reflects only the authors' view; the European Commission and their executive agency are not responsible for any use that may be made of the information the work contains. Computing resources were provided by the North-German Supercomputing Alliance (HLRN).

## Author contributions

C.N., J.H., and M.H. conceived the study. C.N. set up the model simulations, with help from R.T., Ö.G., and J.H. C.N. performed the analysis, created the figures, and wrote the majority of the manuscript. All authors contributed to the interpretation of the results and to finalizing the manuscript.

## Funding

## Competing interests

The authors declare no competing interests.
