## [Peer Review File · Nature Communications]

Abruptly attenuated carbon sequestration with Weddell Sea dense waters by 2100REVIEWER COMMENTS

Reviewer #1 (Remarks to the Author):

This paper by Nissen et al. explores changes in deep ocean carbon sequestration under a future CO₂ emissions scenario. Specifically, they consider how changes in the Weddell Sea - presently an important region of communication between the atmosphere and deep ocean - impact the uptake of carbon below 2000m. The model configuration is exceptionally high-resolution in the region of the Weddell Sea continental shelf and has an explicit representation of ice shelf cavities that are coupled to the ocean. As such, it represents one of the most comprehensive efforts to assess the possible future evolution of the ocean carbon sink in this region.

The paper presents a thorough analysis of the model's evolution, and presents a compelling case for the changing dynamics that evolve over time. I can't fault any of the writing or the presentation of the analysis. Based on the results presented, however, I didn't find that there is sufficient evidence that the pattern observed in the 2090s represents an "abrupt attenuation" of the carbon sequestration (I am inferring from this language, and the assertions of its importance, that the authors are suggesting that this represents a fundamental transition of state). Thus, while I think that the authors have performed and presented their analysis to a high quality, I disagree with the central thesis and motivating narrative of the paper. As such, I can't recommend it for publication.

The root of my disagreement can be traced to Figure 2a. Here, the grey curve shows the evolution of the carbon accumulation below 2000m in the SW Weddell Sea. The central thesis of the paper rests on the assertion that the decreased slope in the final decade of the simulation represents an abrupt attenuation in carbon sequestration. While I agree that the character of that final decade does appear different to much of the preceding century, the whole time-series exhibits substantial decadal-scale variability (likewise in simB). For example, the flattening of the slope in the 2090s would perhaps not be considered substantially distinct to an apparent flattening in the 2030s. It is not beyond doubt, therefore, that what happens in the 2090s represents a fundamental shift in the dynamics of the regional carbon cycle. Put another way, the authors have not shown that what happens in the 2090s represents a statistically significant departure from the system's normal operation, or that normal operation would not proceed again following this perturbation. I think this is probably a prerequisite for a publication making a claim of this sort.

This aspect of the results is obscured in much of the rest of the analysis when decadal averages are considered. Figure 2b emphasizes this, giving the impression that accumulation is increasing over time, between the 1990s, 2000s, and 2080s, before dropping sharply in the 2090s. However, this picture is strongly accentuated by the anomalously large accumulation in the 2080s. Indeed, what happens in the 2080s (as the authors examine) is the best evidence that decadal-scale variability in carbon sequestration can take place in this model, making it all the more challenging to be sure that what happens in the following decade is unequivocally distinct.

The authors acknowledge that the evolution of the system beyond 2100 is not certain. For what

it's worth, I am inclined to agree with them that what happens in the 2090s is indicative of a changing system, since this fits with our conceptual understanding. However, it's not beyond imagination that carbon accumulation might quickly recover, as the basin starts to fill with lighter waters and dense water formation continues again at a consistent pace. If it were feasible to run another few decades of simulation, this would confirm or refute the authors' central assertion. I would welcome this as a revision to the paper, but acknowledge that it is not trivial undertaking. If it is not feasible, I think this remains an interesting, compelling, and insightful analysis of decadal timescale changes in regional carbon sequestration, but not one that merits the more substantive claims of the title and broader narrative.

Sincerely,

Graeme MacGilchrist

Reviewer #2 (Remarks to the Author):

Review of the manuscript "Abruptly attenuated carbon sequestration with Weddell Sea dense waters towards the end of the 21st century" submitted to Nature Communications by Nissen et al.

General comments (overall quality of the manuscript):

The authors evaluated in their manuscript the relation of carbon sequestration with changes on dense water formation in the southern Weddell Sea, through the investigation of two simulations (control and high-CO₂ emission) of a global ocean-ice-biogeochemical model. The modeled results were properly setup and validated, allowing the authors to rise the conclusion that the deep waters of the region decrease their ability to sequester and storage carbon towards the end of the century.

The topic is very timely, interest and relevant for both observers and modelers researchers of the Antarctic and climate community. Also, the manuscript is relevant to improve a better understand of the role of the Weddell Sea on the carbon cycle and ocean dynamics of the Southern Ocean, considering a scenario of high CO₂ levels in the atmosphere.

The manuscript is well written, concise and have relevant (and outstanding) figures to the comprehension of the message. The references are up-to-date. The inclusion of the Supplementary material also helps to the understanding and clarity of the manuscript. It was a pleasure reading. Thus, I think that the manuscript is worth of publication after a round of minor review, focusing on the comments/suggestions below.

Minor comments:

Line 35 – The paragraph is very concise and precise on the information; however, I think that the inclusion of some sentences regarding the distinct varieties of DSW (i.e., HSSW, LSSW and ISW) much be included for clarity of their role on AABW production.

Line 49 – In fact, this is reported since much earlier than the 80s. Please check: Azaneu, M., Kerr, R., Mata, M. M., & Garcia, C. A. (2013). Trends in the deep Southern Ocean (1958–2010): Implications for Antarctic Bottom Water properties and volume export. *Journal of Geophysical Research: Oceans*, 118(9), 4213-4227.

Figure 1 – The authors should acknowledge in the legend what are the studies that previously sketched the water masses circulation both in panels (a) and (b). The authors could indicate in panel (b) the carbon sequestration by AABW (WSDW+WSBW).

I feel missing in the discussion a little bit more about the key role of the increased WDW over the continental shelf acting to change the seawater properties and the ability of the DSW to uptake carbon (since DSW is sourced by the intruded WDW over shelf). For example, is the proportion of DSW (30%) and WDW (70%) that accounts to form AABW (WSDW+WSBW) in the region that will predominantly change (decreasing the ability to ventilate the deep basin and, consequently, to incorporate carbon in the abyss) or the physical and biogeochemical conditions of WDW that upwelled over shelf (enriched with carbon by increased vertical carbon flux; i.e., a changing in the DSW transformation/evolution) that will prevail to reduce the sequester of carbon? A couple of paragraphs on this matter is suggested to be included.

Reviewer #3 (Remarks to the Author):

Review of Abruptly attenuated carbon sequestration with Weddell Sea 2 dense waters towards the end of the 21st century by Nissen et al.

- What are the noteworthy results?

This paper uses a modelling framework to argue that the sequestration of carbon in the Weddell Sea can be subject to rapid changes under future climates, due to mainly physical changes associated with water mass transformations. This is important – the Weddell Sea is a key site globally where dense waters are produced which ultimately flood the global abyss, and the carbon sequestration here exerts a long-term (centennial or longer) impact on planetary-scale climate. I feel the results are noteworthy, and deserve publication – they don't "solve" the problem (and nor do the authors claim that they do), and in particular there are limitations on what can be said associated with computational expense – but they certainly add useful information and will stimulate further work (both modelling and observational) to better understand the key processes and include them in future-generation models.

- Will the work be of significance to the field and related fields? How does it compare to the established literature? If the work is not original, please provide relevant references.

The authors do a good job of positioning their results in the context of existing literature, and drawing out the new findings and their importance. In particular, it is worth noting that recent results by MacGilchrist et al. have enhanced attention on the open-ocean biological system in

the Weddell Sea – combined with horizontal circulation, this is now seen as an important (and previously under-appreciated) vector for carbon drawdown. The present paper does not dispute/overturn this in any way, but demonstrates that predominantly physical processes happening on the shelf (and in ice shelf regions) can have profound and rapid impacts on carbon sequestration. To my mind, this stresses the importance of understanding the cross-disciplinary and cross-shelf-break system of the Weddell Sea in its totality, and incorporating that understanding into models – which the present paper does well. The authors also do a good job of reflecting model advances, from early studies that showed rapid changes in the shelf system were possible due to changing warm water intrusion, to more recent studies that reconcile this apparently more resilience. The model used in the present study aligns better with the latter studies, which is encouraging. Overall, I think the present study adds information that will be useful to climate studies, oceanography, marine productivity, and those seeking optimal routes to progress these issues – it will have broad interest.

- Does the work support the conclusions and claims, or is additional evidence needed?
- Are there any flaws in the data analysis, interpretation and conclusions?
- Do these prohibit publication or require revision?
- Is the methodology sound? Does the work meet the expected standards in your field?

I will answer these questions together...

I believe the conclusions are sound and supported, though (as with any modelling study) there are aspects that could have been explored further, had the computational resource been available. (There is always more that could have been done, right?). So I don't feel additional evidence is needed, but it might be sensible to amend the writing in places to draw out more clearly what the current limitations are, so as to point the way forward for future studies. In particular, the issues that I feel warrant enhanced discussion are:-

- Ensembles. When making projections over several decades or longer, the preferred approach is to use multiple models so that results can be drawn into ensembles and the spread examined as a means of establishing uncertainty. That isn't possible here – the model setup is unique – but the forcing for it seems to come from a single (AWI) climate model. That model is good, but what does its output look like compared to the spread of others?

- Ensembles again. A second good approach is to re-run the model multiple times so as to test how dependant on exact starting conditions the results are. This is quite important when projecting sudden changes at end-of-century, since the timing/magnitude of those might be specific to this run, and be different in runs with subtly different starting conditions/forcings. I realise that computational power is the limiting factor here, but a statement on this limitation and what it means for interpretation of the results would be helpful.

- Scenario. The model uses a high-emissions scenario, one that actually represents higher GHG emissions than we are currently on track for. So arguably the results could represent a case that is worse than we will see, given existing (let alone future) commitments to tackle climate change. It would have been good to include an intermediate scenario and a low-emissions

scenario, for comparison – though again I realise that computational power is the limiting resource. This is discussed a little bit in the text, but I feel more is warranted – in particular, if the authors agree that the likelihood of an abrupt/large-amplitude change is less under lower emissions scenarios, then a statement of that should be included.

- A conceptual point. The model used is a forced-ocean model, albeit it a very clever one with ice shelf cavities etc. Forcing comes from a climate model. This is all fine and well-explained, but does mean that the extra processes resolved in the fine-resolution ocean model cannot feed back onto the large-scale climate that forces it. I doubt this is a particular factor – the likely climate feedbacks will be centennial or longer scales, not decadal scales that the model is run for – but I think it is worth including a statement on this in the paper.

- Is there enough detail provided in the methods for the work to be reproduced?

Yes, but in reality nobody will try to reproduce exactly these results given the complexity of the model and computational resource needed to run it. But I feel that it will stimulate further studies (including by other groups) who will seek to reproduce the basics of this study as part of developing the science further and more fully.

A couple of extra points:-

- The paper highlights the extra presence of WDW on the shelf as a causal factor in the abrupt changes, but does not go into much detail concerning the mechanisms that enable this. (There is some, but not much). I presume these are wind forced changes, reduced sea ice production (and hence shallower mixed layer, less dense shelf water) etc. A little more detail here would be helpful to the reader.

- By comparison, a whole paragraph in the discussion is given over to comparing the performance of the model re AABW export with that in ESMs. I think this can be condensed? (Nobody sensible believes that coarse-resolution ESMs do a good job of AABW formation/export, do they?)

- “Cumulative accumulation” sounds odd to me, for some reason. Why not just “Accumulated...”?

In summary, I feel this is a useful paper that will advance the field and be of interest to quite a broad readership – I recommend publication, subject to revisions as detailed above.

Response to comments by reviewers

Reviewer #1 (Remarks to the Author):

This paper by Nissen et al. explores changes in deep ocean carbon sequestration under a future CO₂ emissions scenario. Specifically, they consider how changes in the Weddell Sea - presently an important region of communication between the atmosphere and deep ocean - impact the uptake of carbon below 2000m. The model configuration is exceptionally high-resolution in the region of the Weddell Sea continental shelf and has an explicit representation of ice shelf cavities that are coupled to the ocean. As such, it represents one of the most comprehensive efforts to assess the possible future evolution of the ocean carbon sink in this region.

The paper presents a thorough analysis of the model's evolution, and presents a compelling case for the changing dynamics that evolve over time. I can't fault any of the writing or the presentation of the analysis. Based on the results presented, however, **I didn't find that there is sufficient evidence that the pattern observed in the 2090s represents an "abrupt attenuation" of the carbon sequestration (I am inferring from this language, and the assertions of its importance, that the authors are suggesting that this represents a fundamental transition of state)**. Thus, while I think that the authors have performed and presented their analysis to a high quality, I disagree with the central thesis and motivating narrative of the paper. As such, I can't recommend it for publication.

The root of my disagreement can be traced to Figure 2a. Here, the grey curve shows the evolution of the carbon accumulation below 2000m in the SW Weddell Sea. The central thesis of the paper rests on the assertion that the decreased slope in the final decade of the simulation represents an abrupt attenuation in carbon sequestration. While I agree that the character of that final decade does appear different to much of the preceding century, the whole time-series exhibits substantial decadal-scale variability (likewise in simB). For example, the flattening of the slope in the 2090s would perhaps not be considered substantially distinct to an apparent flattening in the 2030s. **It is not beyond doubt, therefore, that what happens in the 2090s represents a fundamental shift in the dynamics of the regional carbon cycle. Put another way, the authors have not shown that what happens in the 2090s represents a statistically significant departure from the system's normal operation, or that normal operation would not proceed again following this perturbation.** I think this is probably a prerequisite for a publication making a claim of this sort.

This aspect of the results is obscured in much of the rest of the analysis when decadal averages are considered. Figure 2b emphasizes this, giving the impression that accumulation is increasing over time, between the 1990s, 2000s, and 2080s, before dropping sharply in the 2090s. However, this picture is strongly accentuated by the anomalously large accumulation in the 2080s. Indeed, what happens in the 2080s (as the authors examine) is the best evidence that decadal-scale variability in carbon sequestration can take place in this model, making it all the more challenging to be sure that what happens in the following decade is unequivocally distinct.

The authors acknowledge that the evolution of the system beyond 2100 is not certain. For what it's worth, I am inclined to agree with them that what happens in the 2090s is indicative of a changing system, since this fits with our conceptual understanding. **However, it's not beyond imagination that carbon accumulation might quickly recover, as the basin starts to fill with lighter waters and dense water formation continues again at a consistent pace.** If it were feasible to run another few decades of simulation, this would confirm or refute the authors' central assertion. I would welcome this as a revision to the paper, but acknowledge that it is not trivial undertaking. If it is not feasible, I think this

remains an interesting, compelling, and insightful analysis of decadal timescale changes in regional carbon sequestration, but not one that merits the more substantive claims of the title and broader narrative.

Sincerely,

Graeme MacGilchrist

We thank Graeme MacGilchrist for raising these important points. In the following, we will address his concerns regarding **a)** the decadal variability in deep-ocean carbon accumulation rates and the abruptness of the simulated changes towards the end of the 21st century and **b)** the sustainability of the change beyond the year 2100.

a) Decadal variability in deep-ocean carbon accumulation rates and the abruptness of the simulated changes towards the end of the 21st century: Are the 2090s really different from preceding decades?

We agree with the reviewer that by only having shown selected decades of deep-ocean carbon sequestration rates in the submitted manuscript instead of the full time series, this might have wrongfully suggested to the reader that rates are continuously increasing between the 2000s and the 2080s – which is not the case. In fact, there is substantial decadal and interannual variability in carbon accumulation rates in the southern Weddell Sea, with the 2080s (2090s) displaying the highest (lowest) decadal average rate since the 1990s (see Fig. 1 below, which we have added to the revised manuscript). While the decadal average rate of the 2090s is lower than the mean \pm one standard deviation over all decades, some annual rates within the 2090s are not. Accordingly, while one could conclude from this that the decadal average is indeed different from preceding decades, the rather large interannual variability makes this conclusion, based on the accumulation rates alone, less certain.

In order to statistically assess whether the simulated evolution in any given time series includes abrupt changes that are not within the natural variability of the variable in question, we use the change point analysis (R package “Detection of Structural Changes in Climate and Environment Time Series”; see Beaulieu & Killick 2018, Killick et al., 2021). With this R package, a number of pre-defined linear models is fit to a given time series, and the time series is screened for the existence of change points. Here, following Beaulieu & Killick (2018), we include 8 statistical models in our analysis (a constant mean, a constant linear trend, a constant mean with first-order autocorrelation, a constant linear trend with first-order autocorrelation, multiple change points in the mean, multiple change points in the linear trend, multiple change points in the mean with first-order autocorrelation, and multiple change points in the linear trend with first-order autocorrelation). Subsequently, all model fits are ranked with the Akaike Information Criterion (AIC), which combines the model likelihood with a penalty for the number of parameters fitted (Akaike 1974). Thereby, the statistical model with the lowest AIC score corresponds to the model that is best suited to describe the time series in question.

Applying this methodology to the simulated deep-ocean carbon sequestration rates between 1980-2100 in *simA*, the tool reveals three change points in the linear trend for the second half of the 21st century (Fig. 1c). While two change points mark the start (2073) and end (2084) of the high-accumulation phase in the late 2070s and the 2080s, the sign of the linear trend already changes from positive to negative in the year 2050. This implies that the high rates preceding the 2090s are a temporary phenomenon, interrupting the general trend

towards lower accumulation rates, and suggesting that the attenuated rates in the last decade are the result of an underlying system change.

In general, to what extent such a systematic change over the 21st century is controlled by changes in atmospheric CO₂ levels or changes in climate can be assessed by comparing our model experiment *simA* (historical + SSP5-8.5 scenario, varying atmospheric CO₂ levels and varying climate) to a simulation in which only atmospheric CO₂ levels vary, but not climate (*simC*; we have added this simulation to the revised manuscript). In *simC*, deep-ocean carbon accumulation rates are overall much less variable than in *simA*, but are higher towards the end of the 21st century than in the 1990s (Fig. 1b, d & f). Notably, the accumulation rates increase after 2084 in *simC*, when the rates decline in *simA*. Altogether, this implies that climate-induced changes explain the attenuated accumulation rates at the end of the 21st century in *simA*, outweighing the increase in accumulation expected from increased atmospheric CO₂ levels alone. This lends support to the finding that the physical environment in the Weddell Sea at the end of the 21st century is indeed different from earlier in the simulation, with subsequent changes in deep-ocean carbon accumulation.

In fact, in line with the decline in bottom water ventilation along the continental slope (see Fig. 4 of the revised manuscript), the difference in density between bottom waters on the southern continental shelf and the deep ocean, which constitutes the main pathway for newly formed dense waters into the abyss (see Fig. 1 in the revised manuscript), is higher in the 2090s than it was in the 1990s (Fig. 2 below). In particular, the rate of decoupling accelerates after the early 2070s (change point in 2072 in the linear trend + first-order autocorrelation; Fig. 2b), demonstrating how the deep ocean is effectively cut off from the continental shelf sea towards the end of the 21st century, when bottom waters from the southern continental shelves are too light to reach the deep ocean of the central southern Weddell Sea.

With changes in the physical environment starting already before the 2090s, the high-mixing event in the 2080s only temporarily masks the trend towards lower deep-ocean carbon transfer with dense waters in the southern Weddell Sea. In fact, without open-ocean mixing, total carbon accumulation between 2080 and 2100 would be 59% lower (Fig. 1a, c & e; note that model output of the physical fluxes is unfortunately not available for the period between 2010-2079, so that the impact of vertical mixing on the deep-ocean carbon budget cannot be quantified for this time period). For this time period, the residual average deep-ocean carbon accumulation rate (3.14 Tg C yr⁻¹) can largely be explained by biological fluxes (90%), further demonstrating the low deep-ocean carbon transfer with dense waters by the end of the 21st century. In agreement, vertical volume fluxes are increasingly directed upwards especially along the southwestern slopes towards the end of the 21st century (Fig. 3 below).

Altogether, this suggests that the physical environment at the end of the 21st century is markedly different from the one at the end of the 20th century or even the beginning of the 21st century. As a direct consequence, the changes in the downward transfer of carbon with Weddell Sea dense waters along the continental slopes can be interpreted as significant. In the revised manuscript, we have reworked the result and discussion sections to better reflect this key message (see below).

Fig. 1: New Fig. 2 in the revised manuscript: Deep-ocean carbon accumulation. **a, b** Accumulated carbon in Pg C between 1980 and 2100 below 2000 m in the southern Weddell Sea south of the WOCE transect SR4 (see blue area in inlet) in **a** the model simulation *simA* (dark grey; historical + SSP5-8.5 scenario) and **b** the control simulation *simB* (dashed dark grey) and *simC* (solid dark grey; constant climate + varying atmospheric CO₂). Panels **c-f** show carbon accumulation rates in Tg C yr⁻¹ averaged **c, d** annually and **e, f** for each decade in **c, e** *simA* and **d, f** *simC*. In panels **c, d**, the dark blue lines indicate the statistical models providing the best fits to describe the time series, and change points are indicated with vertical lines and as squares on the x axis (see Methods and references^{61,62}). In panels **e, f**, the whiskers depict one standard deviation within each decade, the horizontal blue line indicates the average accumulation rate between 1990-2100, and the shading shows ± one standard deviation around the mean. The light grey lines in panels **a, c** and the light grey bars in panel **e** denote the accumulated carbon due to all processes except vertical mixing in the open ocean north of the 3500 m isobath (dotted line in the map in panel **a**). The contribution of biological and physical fluxes (the latter calculated as residual) in **a, g** *simA* and **b, h** *simC* are shown in green and blue, respectively. Biological fluxes are dominated by sinking particle fluxes (Supplementary Fig. 2).

Fig. 2: This Figure was added as Fig. 5 to the revised manuscript: Evolution of bottom density between 1980 and 2100 in the southern Weddell Sea. Panel a shows the average air temperature in $^{\circ}C$ over the southern Weddell Sea (see map) used to force *simA* (historical + SSP5-8.5). Panel b shows the difference in bottom density (σ_2 , i.e., potential density referenced to 2000 dbar) in $kg\ m^{-3}$ between the open ocean and two locations on the continental shelf (light and dark blue, respectively) in *simA* (thick lines) and in the control simulation *simB* (thin lines). See symbols in the map for the exact locations used to compute the density differences. For *simA*, the straight lines indicate the statistical models providing the best fits to describe the time series, and change points are indicated with vertical lines and as squares on the x axis (see Methods and references^{61,62}). Panel c indicates the corresponding evolution of bottom density along the whole transect in the southern Weddell Sea (see map) in *simA* (colors) and the bottom topography along this transect (black line). The location of the shelf break, defined as the 700 m isobath, is denoted by the horizontal grey line.

Fig. 3: Vertical volume transport in the southern Weddell Sea. Vertical volume transport across 2000 m in Sverdrup (Sv; $1 \text{ Sv} = 1 \cdot 10^6 \text{ m}^3 \text{ s}^{-1}$) south of the WOCE transect SR4 in the model simulation *simA* (solid lines; historical + SSP5-8.5 scenario) and the control simulation *simB* (dotted lines). Fluxes are integrated over **a** the continental slope (dark blue) and the open ocean (light blue), which are separated at the 3500 m isobath (see inlet), and **b** the continental slope west (black) and east (light grey) of 30°W. After the year 2100, thick and thin lines correspond to the extensions *ext1* and *ext2*, respectively (see Methods). This Figure was added as Fig. 6 to the revised supplementary material.

In response to this comment by reviewer #1, we have added all the plots above either as revised Figures in the main part of the manuscript or to the revised Supplementary material (see respective Figure captions). Further, we have reworked all bar plots in the revised manuscript, in order to make clearer that only selected decades are shown (see Fig. 3, 6 & 7 in the revised manuscript). In addition, we have made a number of changes in the main text, as summarized in the following:

In the revised method section, we have included a description of 1) *simC* (constant climate, but varying atmospheric CO_2) and 2) the change point analysis:

1) “Lastly, *simC* is forced with varying atmospheric $p\text{CO}_2$ levels (as in *simA*), but constant climate (as in *simB*). With this simulation, changes in deep-ocean carbon accumulation rates resulting from rising atmospheric $p\text{CO}_2$ alone, i.e., without the concurrent impact of a changing climate, can be quantified.”

2) “**Change Point Analysis.** In order to statistically identify (abrupt) changes in the simulated evolution of physical or biogeochemical quantities of the southern Weddell Sea over the 21st century, we use the R package “Detection of Structural Changes in Climate and Environment Time Series” (“EnvCPT”, see references^{61,62}). With this package, a number of pre-defined linear models is fit to a given time series, and the time series is screened for the existence of

change points. Here, following the reference⁶¹, we include 8 statistical models in our analysis: a constant mean, a constant linear trend, a constant mean with first-order autocorrelation, a constant linear trend with first-order autocorrelation, multiple change points in the mean, multiple change points in the linear trend, multiple change points in the mean with first-order autocorrelation, and multiple change points in the linear trend with first-order autocorrelation. Subsequently, all model fits are ranked with the Akaike Information Criterion (AIC), which combines the maximum likelihood estimate for each statistical model, i.e., the optimal parameter set to describe the time series at hand, with a penalty for the number of parameters fitted⁶⁵. Thereby, the statistical model with the lowest AIC score corresponds to the model that is best suited to describe the time series in question. We apply this methodology to the time series between 1980-2100 of annual mean deep-ocean carbon accumulation rates in *simA* and *simC* (Fig. 2c & d), air-sea CO₂ exchange in *simA* (Supplementary Fig. 1), and the difference in bottom density between the continental shelf and the open ocean in the southern Weddell Sea in *simA* (Fig. 5). For each of these quantities, the model including multiple change points in the linear trend is the best to describe the time series (with first-order autocorrelation for the dark blue line in Fig. 5 and without for all others). As we are mostly interested in decadal changes of annual mean properties in this study, we use a minimum segment length of 10 in the analysis, i.e., there has to be at least 10 years in between any two change points identified. We note, however, that the exact outcome of the change point analysis, including the timing of any change points, might be sensitive to this choice⁶¹. In particular, while the identified change points in the linear trend of deep-ocean carbon accumulation rates in *simA* (see Fig. 2c) are unaffected by choosing a minimum segment length <10, they change if larger intervals are chosen (e.g., change points in the mean with first-order autocorrelation are identified in 2020 and 2076 for a segment length of 12 or 15). This sensitivity can be understood by the length of the high-accumulation phase in the late 2070s and the 2080s of approximately 10 years, which prevents the identification of the onset and end of this event as individual change points if the required minimum segment length is >10.”

In the first part of the result section, we have added a paragraph discussing the identified change points and the newly added model experiment *simC*. It reads:

“Even though the deep-ocean carbon accumulation rate in the 2090s in *simA* is lower than the average (\pm one standard deviation) over all decades (Fig. 2e), both decadal and interannual variability complicate the assessment whether the attenuation at the end of the 21st century is statistically significant (Fig. 2c). To resolve this, we apply the change point analysis, which uncovers any shifts in the mean or the linear trend of a given quantity that exceed natural variability (see Methods and references^{61,62}). For the second half of the 21st century, this analysis reveals three change points in the linear trend in annual mean deep-ocean carbon accumulation rates (dark blue squares in Fig. 2c). While two change points mark the start (2073) and end (2084) of the high-accumulation phase in the late 2070s and the 2080s, the trend in the accumulation rate switches from positive to negative in the year 2050 (dark blue lines in Fig. 2c), as indicated by another change point at that time. This suggests that the reduced rates in the 2090s are the result of an underlying system change. Further, this implies that the high rates in the 2080s are a temporary phenomenon, interrupting the general trend towards lower accumulation rates. To what extent the trends or variability in deep-ocean carbon accumulation rates over the 21st century are controlled by changes in atmospheric CO₂ levels or other climate-related processes can be assessed by comparing *simA* to a simulation in which only atmospheric CO₂ levels vary, but not climate (*simC*; see Methods). In *simC*, deep-ocean carbon accumulation rates are overall much less variable than in *simA*, but are higher towards the end of the 21st century than in the 1990s (Fig. 2b, d & f). Notably, the accumulation rates increase after 2084 in *simC*, when the rates decline in *simA*. Altogether, this implies that climate-induced changes in carbon sequestration processes lead to most of the decadal and interannual variability in *simA* and explain the attenuated accumulation rates at the end of the 21st century, outweighing the

increase in accumulation expected from increased atmospheric CO₂ levels alone. Still, this analysis leaves it unclear whether the overall reduction in simA after the year 2050 and the intermittent increase are caused by changes in the same mechanism or by the superposition of two mechanisms with different underlying trends or variability.”

In the second part of the result section, in order to put more emphasis on our main take-away message, we have included a more thorough description of the importance of simulated changes along the continental slope (which constitute the pathway for the deep-ocean transfer of carbon with Weddell Sea dense waters) and the role of open-ocean mixing:

“In general, towards the late 21st century, an increasing amount of carbon is transferred to the deep ocean for any given open-ocean mixing event, as carbon concentrations in the upper ocean exceed those in the deep ocean after the 2060s (reddish colored vertical profiles in Fig. 3d). In fact, without open-ocean mixing, total deep-ocean carbon accumulation between 2080 and 2100 would be 59% lower (light grey lines in Fig 2a & c). The residual average carbon accumulation rate between 2080-2100 (3.1 Tg C yr⁻¹) can largely be explained by biological fluxes (90%), implying continuously low deep-ocean carbon transfer between 2080-2100 with Weddell Sea dense waters. Irrespective of that, open-ocean mixing events represent positive anomalies in deep-ocean carbon accumulation, temporarily overriding the underlying shift towards a reduced downward carbon transfer with dense waters along the southwestern continental slopes in the Weddell Sea (Fig. 2c). For example, the enhanced open-ocean downward flux of carbon in the 2080s is the result of the downward mixing of relatively recently ventilated and thus carbon-enriched dense waters originating from regions upstream of the southern Weddell Sea (Supplementary Fig. 6 & 7 and Supplementary Section 2). Overall, the dominance of vertical physical fluxes in controlling the variability in carbon transfer to depth - in particular the attenuated transfer in the 2090s - implies that deep-ocean carbon accumulation rates are sensitive to physical processes in the overlying water column and upstream on the Weddell Sea continental shelves.”

b) Sustainability of the change: What will happen after the year 2100?

In response to the reviewer’s comment, we have extended our model experiment *simA* (historical + SSP5-8.5) by another 50 years until the year 2150. However, as the scenario forcing from the AWI Climate Model is not available after the year 2100 in the high-emission scenario considered here (Semmler et al., 2020), the extension was forced with increasing atmospheric pCO₂ levels (reaching 1737 ppm in the year 2150), but repeat-year forcing for all other atmospheric variables. In particular, we ran two 50-year-long extensions, which are forced by repeating the years 2091 (*ext1*) and 2095 (*ext2*), respectively. These years were again identified by assessing the state of the Southern Annular Mode and El Niño Southern Oscillation – this time aiming to identify the two most “normal” years in the 2090s (following the procedure for the control experiment *simB*, see Methods).

In the absence of forcing fields after the year 2100, these two extensions have to be considered idealized cases for which warming comes to a standstill. Even more so, Weddell Sea air temperatures in *ext2* (-8.4°C) are lower than in *ext1* (-7.6°C) and lower than the decadal mean temperature for both the 2090s (-7.4°C) and the 2080s (-8.1°C). This demonstrates the considerable interannual variability in air temperatures also in the 2090s, which superimposes the overall trend towards increasing air temperatures (see Fig. 5 below). Thereby, while *ext1* can be considered a continuation of average conditions in the 2090s, *ext2* should be considered a “cold scenario”, in which the surface layers in the Weddell Sea are exposed to colder air temperatures in the extension than towards the very end of the 21st century.

We find that in *ext1*, deep-ocean carbon accumulation in the southern Weddell Sea stays at the low rates of the 2090s for another ~25 years, before another open-ocean mixing event temporarily increases the accumulation rates (Fig. 4 below). In *ext2*, these high-accumulation events appear more frequent, so that the total additional accumulation between 2100 and 2150 in *ext2* (0.58 Pg C) is almost twice as high as in *ext1* (0.32 Pg C). Similar to the high-accumulation event in the 2080s, episodically enhanced open-ocean mixing leads to the episodically enhanced downward flux of carbon in both extensions (Fig. 6).

In general, as more carbon accumulates in the upper ocean over the 21st century and in the model extensions, any open-ocean mixing event leaves an increasingly large imprint on the deep-ocean carbon budget over time, as in the 2060s, the sign of the vertical gradient of total carbon concentrations in the southern Weddell Sea changes, and carbon concentrations are higher in the upper ocean than in the deep ocean (vertical profiles in Fig. 7d). Unfortunately, an evaluation of both the frequency and magnitude of such events for the present-day is prevented by the lack of adequate observations, leaving it unclear whether the simulated events in our model are realistic. In our model, these open-ocean mixing events are possibly more frequent in *ext2* because surface waters in this model extension are exposed to relatively cold air temperatures, favoring the occurrence of vertical instabilities (see vertical diffusivity in Fig. 6d) and resulting in the simulated large imprint of open-ocean mixing on the deep-ocean carbon budget of the southern Weddell Sea. In fact, without open-ocean mixing, total carbon accumulation between 2080 and 2150 would be 52% (*ext1*) and 55% (*ext2*) lower (see Fig. 2). Still, as 88% of the residual average deep-ocean carbon accumulation rate of 3.2 Tg C yr⁻¹ (*ext1*) can be explained by biological fluxes, this implies sustained low deep-ocean carbon transfer with Weddell Sea dense waters also after the year 2100 – even under the halted warming assumed here.

In support of this, the decoupling of the continental shelf sea and the deep ocean is sustained until the end of the model extensions (Fig. 5 & 8). In particular, the density difference of bottom waters on the southern continental shelves and the deep ocean offshore stabilizes at 0.37 kg m⁻³ (*ext1*) and 0.33 kg m⁻³ (*ext2*), respectively (Fig. 5). Even though this is lower than the difference in the 2090s (0.44 kg m⁻³), this should not be interpreted as a reversal of the change, but rather as a consequence of the setup of the model extensions (no further warming after 2100), given that the density difference at the end of the extensions is still ~three times larger than the one in the 1990s. Similarly, also the rather small changes in bottom water properties and sea ice formation on the southern continental shelves at the end of *ext1* can likely be attributed to the absence of further warming in the atmosphere, with ice-shelf basal melt rates displaying a less direct dependence on air temperatures, increasing by another 23% by the end of *ext1* (278 Gt yr⁻¹; relative to 223 Gt yr⁻¹ the 2090s; Fig. 8).

Overall, given that the conditions in the 2090s are largely sustained until the end of *ext1*, our findings suggest that no quick re-filling of the deep ocean with the lighter waters from the shelf can be expected after the year 2100. Even more so, given that we can draw this conclusion even under halted warming, a quick reinvigoration of deep-ocean carbon transfer along the continental slopes can even less be expected if warming continued after 2100, which would be expected from the substantial rise of atmospheric CO₂ under the SSP5-8.5 scenario between the years 2100 (1135 ppm) and 2150 (1737 ppm). In reality, under continued atmospheric warming, the freshening and warming of shelf waters would very likely continue, further decoupling bottom waters on the shelf and in the deep ocean. This is also supported by the fact that the decoupling of bottom waters between the continental shelf sea and the deep ocean accelerates in the 2070s (as indicated by a change point in the first half of this decade, see Fig. 2 above), implying that the deep ocean would remain cut off from bottom water renewal with waters descending the slope for years and decades to come.

Further support for this can be drawn from paleoceanographic literature, which suggests that re-establishing the pathway of dense waters from the high-latitude continental shelves to the abyss, which got cut off due to enhanced freshwater runoff in coastal regions in response to warming, takes multiple centuries or more (see e.g., Silvano et al., 2018, Thomas et al., 2020, Turney et al., 2020). Altogether, while it remains unresolved to what extent open-ocean mixing might accelerate the filling of the deep basin of the Weddell Sea with the lighter waters, our model experiments clearly show that on decadal time scales, a reinvigoration of the deep-ocean carbon transfer with Weddell Sea dense waters is unlikely.

Fig. 4: Deep-ocean carbon accumulation in the model extensions. **a** Accumulated carbon in Pg C between 1980 and 2100 below 2000 m in the southern Weddell Sea south of the WOCE transect SR4 (see blue area in inlet) in the model simulation *simA* (dark grey; historical + SSP5-8.5 scenario) and for the 50 years of the model extensions *ext1* (thick dark grey) and *ext2* (thin dark grey). The contribution of biological and physical fluxes (the latter calculated as residual) of *simA+ext1* is shown in green and blue, respectively. **b** Annual carbon accumulation rates in Tg C yr⁻¹ for *simA* (dark grey) and the model extensions *ext1* (thick dark grey) and *ext2* (thin dark grey). As in Fig. 2 of the main manuscript, the dark blue lines indicate the statistical models providing the best fits to describe the time series, and change points are indicated with vertical lines and as squares on the x axis (see Methods and references^{1,2}). The light grey lines in both panels denote the accumulated carbon due to all processes except vertical mixing in the open ocean north of the 3500 m isobath (dotted line in the map in panel **a**). Note that due to their setup, the time axis of the model extensions corresponds to simulation years rather than calendar years. See Method section of the main manuscript for details. This Figure was added as Fig. 13 to the revised supplementary material.

Fig. 5: Evolution of air temperatures and bottom density in the model extensions. Panel **a** shows the average air temperature in °C over the southern Weddell Sea (see map) used to force *simA* between 1980 and 2100 (historical + SSP5-8.5). After the year 2100, the model experiment is extended for another 50 years by forcing it with constant air temperatures as *ext1* (thick line) and *ext2* (thin line), respectively. Panel **b** shows the difference in bottom density (σ_2 , i.e., potential density referenced to 2000 dbar) in kg m^{-3} between the open ocean and two locations on the continental shelf (light and dark blue, respectively) between 1980 and 2100 in *simA* and in the model extensions *ext1* (thick lines) and *ext2* (thin lines). See symbols in the map for the exact locations used to compute the density differences. The straight lines indicate the statistical models providing the best fits to describe the time series, and change points are indicated with vertical lines and as squares on the x axis (see Methods and references^{1,2}). Note that due to their setup, the time axis of the model extensions corresponds to simulation years rather than calendar years. See Method section of the main manuscript for details. This Figure was added as Fig. 14 to the revised supplementary material.

Fig. 6: Divergence of physical flux components in the model extensions and vertical diffusivity in the southern Weddell Sea. Changes in the total carbon inventory south of WOCE transect SR4 and below 2000 m in the model extensions *ext1* and *ext2* due to **a** vertical mixing across 2000 m, **b** vertical advection across 2000 m, and **c** lateral advection across the transect SR4 and the sum of all other flux components, e.g., from the eddy parametrization (dashed bars). All fluxes are in Tg C yr^{-1} , but note that the fluxes in panel **c** are one order of magnitude smaller than those in panels **a**, **b**. Positive fluxes denote an

increase in the deep-ocean carbon inventory in the volume of interest due to the respective flux component. Panel **d** shows the average vertical diffusivity in $\text{m}^2 \text{s}^{-1}$ at 2000 m in the open ocean (see inlet) in *simA* (black; historical + SSP5-8.5 scenario) and the control simulation *simB* (grey). After the year 2100, thick and thin lines correspond to the model extensions *ext1* and *ext2*, respectively. Note that due to their setup, the time axis of the model extensions corresponds to simulation years rather than calendar years. See Method section of the main manuscript for details. This Figure was added as Fig. 15 to the revised supplementary material.

Fig. 7: New Fig. 3 in the revised manuscript Divergence of physical flux components contributing to changes in the deep-ocean carbon inventory. Changes in the total carbon inventory south of WOCE transect SR4 and below 2000 m in the model simulation *simA* (historical + SSP5-8.5 scenario) in the 1990s, 2000s, 2080s, and 2090s due to **a** vertical mixing across 2000 m, **b** vertical advection across 2000 m, and **c** lateral advection across the transect SR4 and the sum of all other flux components, e.g., from the eddy parametrization (dashed bars). Positive fluxes denote an increase in the inventory in the volume of interest due to the respective flux component. All fluxes are in Tg C yr^{-1} , but note that the fluxes in panel **c** are one order of magnitude smaller than those in panels **a**, **b**. Panel **d** shows vertical profiles of total carbon concentrations in mmol m^{-3} averaged over each decade from 1990-2100 in *simA*. Profiles are colored as a function of time in shades of blue and red depending on whether concentrations at 100 m are lower (blue) or higher (red) than those at 2000 m, respectively. Vertical fluxes in panels **a**, **b** and vertical profiles in panel **d** are shown for the continental slope and the open ocean, which are separated at the 3500 m isobath (see inlet in panels **a**, **b**).

Fig. 8: Figure S16: Changes in sea ice growth, basal melt, and water mass properties in the model extensions. **a** Sea-ice growth in m yr^{-1} , **b** ice-shelf basal melt rates in Gt yr^{-1} , **c** bottom potential density anomaly (σ_2 in kg m^{-3} , i.e., potential density referenced to 2000 dbar minus 1000 kg m^{-3}), **d** bottom salinity, **e** bottom potential temperature in $^{\circ}\text{C}$, and **f** total heat content in 10^{21} J in the 1990s and 2090s in *simA* and the final decade of the model extensions *ext1* and *ext2*. All quantities are averaged for the southern continental shelf (blue area in the map). **g** Change in the distribution of σ_2 at a transect along the continental shelf break (see map) between the last decade in *ext1* and the 1990s in *simA*. Panels **h**, **i** show the change between the last decade in *ext1* and the 1990s in *simA* in **h** the age-tracer-based fraction of water that is ventilated at the bottom (middle) and across the continental slope in the western (SR4 west; left) and southern Weddell Sea (WS south; right) and **i** the distribution of potential density anomalies (σ_2 in kg m^{-3}) at the surface, 700 m, and 2000 m. Hatching in panels **g-i** denotes the presence of Weddell Sea Deep Water and Weddell Sea Bottom Water in the last decade of *ext1*, defined in the model as waters with a potential temperature $< -0.2^{\circ}\text{C}$ and a practical salinity > 34.55 . This Figure was added as Fig. 16 to the revised supplementary material.

In response to this comment by reviewer #1, we have included the model extensions *ext1* and *ext2* in the Supplementary material, in the method section, and in some locations of the main text of the revised manuscript. Accordingly, we have made a number of changes in the text, which will be summarized in the following:

In the revised method section, we have included a description of the two model extensions. It now reads:

“Further, given that carbon sequestration rates are attenuated towards the end of the 21st century in simA, this model experiment was extended by another 50 years in order to determine whether the reduced rates in the 2090s a) represent a lasting change in the downward transfer of carbon with Weddell Sea dense waters or b) can be explained by

decadal variability. However, as atmospheric output from the AWI-CM is not available beyond the year 2100, the extension is forced with further increasing atmospheric $p\text{CO}_2$ levels according to the SSP5-8.5 scenario⁸¹, but with constant climate forcing. We run two 50-year-long extensions which are forced by repeating the years 2091 (ext1) and 2095 (ext2), respectively. These two years were again chosen based on the state of the Southern Annular Mode and El Niño Southern Oscillation (see methodology for simB), in this case aiming to represent "normal" atmospheric conditions in the 2090s. Due to this experimental design, the simulated evolution of carbon sequestration rates for the 50 years in ext1 and ext2 constitute idealized cases for the considered high-emission scenario, as atmospheric warming would continue after the year 2100 rather than a temperature stabilization as assumed here under continuously increasing atmospheric $p\text{CO}_2$ levels. While being $>5^\circ\text{C}$ higher than in the 1990s (-13.8°C), the annual mean air temperature in the southern Weddell Sea in ext2 (-8.4°C) is lower than the one in ext1 (-7.6°C) and also than the decadal mean temperature for both the 2090s (-7.4°C) and the 2080s (-8.1°C), demonstrating the considerable interannual variability in air temperatures also in the 2090s, which superimposes the overall trend towards increasing air temperatures (Fig. 5a). As a result, Weddell Sea surface waters experience an overall lower air temperature in ext2 than towards the end of simA, explaining the more frequent occurrence of vertical instabilities, which lead to enhanced deep-ocean carbon accumulation via open-ocean mixing (see Supplementary Section 1).“

In addition, we have added a description of the results of the model extensions to the revised Supplementary material (Section 1):

“As described in more detail in the Method section, atmospheric CO_2 levels continue to rise throughout both model extensions (ext1 and ext2), whereas both extensions were run under the assumption that the atmospheric temperature stabilizes (at two different constant near-end-of-century levels; Fig. S14). As a consequence of the rise in atmospheric CO_2 levels, the oceanic uptake of CO_2 in the southern Weddell Sea increases to 85 Tg C yr^{-1} by the end of the model extensions (Fig. S1a), and the upper ocean carbon inventory continues to rise accordingly (Fig. S1b). At the same time, following the decade of lowest deep-ocean carbon accumulation in the southern Weddell Sea in simA, carbon accumulation stays low for the first two decades in ext1, before another high-accumulation phase occurs (thick dark grey line in Fig. S13). In ext2, which is forced with lower atmospheric temperatures than ext1 (Fig. S14a), these high-accumulation events are more frequent (thin dark grey line in Fig. S13), leading to almost twice as much additional accumulation after 50 years in ext2 (0.58 Pg C) than in ext1 (0.32 Pg C). In agreement with simA, the variability of carbon accumulation rates in the deep ocean of the southern Weddell Sea in both extensions are mainly controlled by physical fluxes (blue line in Fig. S13a), with open-ocean mixing being the flux component controlling the high-accumulation events (Fig. S15). In fact, without open-ocean mixing, total deep-ocean carbon accumulation between 2080 and the end of the model extensions would be 52% and 55% lower for ext1 and ext2, respectively (light grey lines in Fig. S13). Taking ext1 as an example, also the residual average carbon accumulation rate between 2080 and year 50 of this extension (3.2 Tg C yr^{-1}) can largely be explained by biological fluxes (88%), implying that the low deep-ocean carbon transfer with Weddell Sea dense waters towards the end of simA is sustained also in this model extension at stabilized near-end-of-century atmospheric temperatures.

Towards the end of the 21st century in simA, the lower density of newly formed dense waters on the continental shelves prevents these from reaching the abyss in the open-ocean Weddell Sea. Overall, the simulated changes in water mass properties on the southern continental shelves are sustained throughout the model extensions (especially for ext1, see Fig. S16 & Fig. S16). In ext1, sea ice formation stabilizes at a rate close to that projected for the 2090s (Fig. S16a; possibly a consequence of the constant atmospheric forcing applied in the extension), whereas basal melting has further increased to 120% of that in the 1990s by the end of the extension (Fig. S16b; possibly in response to the further increase in the heat content of waters over the shelf, see Fig. S16f). As a consequence, the decoupling between

bottom waters on the southern continental shelves and those in the open ocean is sustained in both model extensions. After 2100, the density difference stabilizes at 0.37 kg m⁻³ in ext1 and 0.33 kg m⁻³ in ext2. While this is lower than the density difference in the 2090s (0.44 kg m⁻³), a stabilization is expected due the setup of the model extensions (no further warming after 2100; Fig. S14a) and should therefore not be interpreted as a reversal of the simulated decoupling between the continental shelf sea and the deep ocean over the 21st century (see Methods). More importantly, given that the density difference is still at least three times larger by the end of the model extensions than in the 1990s, the sustained reduced connectivity implies that more than a few decades and possibly also a larger reversal of the end-of-century warming than applied in ext2 are needed to re-establish the pathway for newly formed dense waters from the continental shelf sea to the abyss of the southern Weddell Sea.”

In the revised discussion section, we have added references to these new results in multiple locations:

“Notably, the decoupling of bottom waters on the southern continental shelves and the open ocean is sustained in two idealized 50-year-long model extensions, which were forced with constant near-end-of-century air temperatures (ext1 and ext2; see Supplementary Section 1, Supplementary Fig. 13-16, and Methods).”

“Since the basal melt rate increases further in our model extension even under constant end-of-century temperatures (up to 288 Gt yr⁻¹ after 50 years in ext1; see Supplementary Fig. 16), basal melt rates can be expected to rise fast beyond 2100 under continued atmospheric warming, further freshening the shelf and subsequently reducing the density of newly formed dense waters and resulting in a continued low transfer of carbon to the abyss of the southern Weddell Sea.”

“While our two model extensions under constant end-of-century atmospheric forcing demonstrate that the evolution of carbon sequestration beyond 2100 is sensitive to the evolution of air temperatures, the decoupling between the continental shelf sea and the deep ocean is sustained for another 50 years even under the lower air temperatures in ext2 (Supplementary Section 1). This demonstrates that more than a few decades and possibly also a larger reversal of the end-of-century warming than applied in ext2 are needed to re-establish the pathway for dense waters from the continental shelves to the abyss in the southern Weddell Sea, which is in line with paleoceanographic literature⁷³⁻⁷⁷.”

Reviewer #2 (Remarks to the Author):

General comments (overall quality of the manuscript):

The authors evaluated in their manuscript the relation of carbon sequestration with changes on dense water formation in the southern Weddell Sea, through the investigation of two simulations (control and high-CO₂ emission) of a global ocean-ice-biogeochemical model. The modeled results were properly setup and validated, allowing the authors to rise the conclusion that the deep waters of the region decrease their ability to sequester and storage carbon towards the end of the century.

The topic is very timely, interest and relevant for both observers and modelers researchers of the Antarctic and climate community. Also, the manuscript is relevant to improve a better understand of the role of the Weddell Sea on the carbon cycle and ocean dynamics of the Southern Ocean, considering a scenario of high CO₂ levels in the atmosphere.

The manuscript is well written, concise and have relevant (and outstanding) figures to the comprehension of the message. The references are up-to-date. The inclusion of the Supplementary material also helps to the understanding and clarity of the manuscript. It was

a pleasure reading. Thus, I think that the manuscript is worth of publication after a round of minor review, focusing on the comments/suggestions below.

We thank the reviewer #2 for this positive feedback. We have included all his/her suggestions in the revised manuscript, as outlined below.

Minor comments:

- 1) Line 35 – The paragraph is very concise and precise on the information; however, I think that the inclusion of some sentences regarding the distinct varieties of DSW (i.e., HSSW, LSSW and ISW) much be included for clarity of their role on AABW production.

We have added some information on the varieties of Dense Shelf Water in the revised manuscript. The paragraph now reads:

“Generally, the formation of AABW can be divided into two step¹³⁻¹⁶: First, open-ocean water masses flowing into the southern Weddell Sea on the eastern flank of the cyclonic Weddell Gyre^{17,18} are transformed to Dense Shelf Water (DSW) as a result of the buoyancy loss caused by atmosphere–ocean, sea ice–ocean, and ice shelf–ocean interactions¹⁴ (Fig. 1). The densification of waters thereby predominantly occurs on the southwestern Weddell Sea continental shelf as a result of heat loss and local sea-ice formation; it is to some extent counteracted by precipitation and meltwater fluxes from the Filchner-Ronne and Larsen ice shelves¹⁹⁻²⁴. In particular, the densest variety of DSW, i.e., High Salinity Shelf Water, which is formed due to cooling and brine rejection in sea-ice formation, is modified in the ice-shelf cavity to form the less saline, but colder Ice Shelf Water, which then mixes with the Low Salinity Shelf Water mostly prevailing on the southeastern Weddell Sea continental shelf^{15,19}. Second, this resulting DSW cascades down the continental slope forming either Weddell Sea Deep Water (WSDW¹⁷; potential temperature $-0.7^{\circ}\text{C} \leq \theta_0 < 0^{\circ}\text{C}$) or Weddell Sea Bottom Water (WSBW¹⁷; $\theta_0 < -0.7^{\circ}\text{C}$). Along the way, Warm Deep Water (WDW¹⁷; the Weddell Sea variant of Circumpolar Deep Water (CDW) with $\theta_0 > 0^{\circ}\text{C}$) is entrained to ultimately form AABW, which then spreads throughout the abyss of the global ocean^{13,14}. Integrating observations from 1973-2017, it has recently been suggested that, on average, 4.5 ± 0.3 Sv of DSW formed on the Weddell Sea continental shelf in the first step entrain another 3.9 ± 0.5 Sv of WDW on its way to the abyss, resulting in a total of 8.4 ± 0.7 Sv of AABW transported northwards along the western flank of the Weddell Gyre¹⁶.”

- 2) Line 49 – In fact, this is reported since much earlier than the 80s. Please check: Azaneu, M., Kerr, R., Mata, M. M., & Garcia, C. A. (2013). Trends in the deep Southern Ocean (1958–2010): Implications for Antarctic Bottom Water properties and volume export. *Journal of Geophysical Research: Oceans*, 118(9), 4213-4227. We thank the reviewer for pointing us to this reference, which we were not aware of. In the revised manuscript, we have added this reference to the introduction and have adapted the statement accordingly. It now reads:

“Both WSDW and WSBW have been warming and freshening since the late 1950s²⁸⁻³⁴, which has at least partly been attributed to property changes of newly formed DSW^{26,35}, possibly due to changes in sea-ice formation and ice-shelf basal melt rates in the area³³.”

- 3) Figure 1 – The authors should acknowledge in the legend what are the studies that previously sketched the water masses circulation both in panels (a) and (b). The authors could indicate in panel (b) the carbon sequestration by AABW (WSDW+WSBW).

As suggested, we have added the references Marshall & Speer (2012) and Vernet et al. (2019) for the sketched circulation to the revised Figure caption. It now reads:

“Figure 1: Sketch illustrating the major processes involved. **a** Water depth in m below the ocean surface in the Weddell Sea. Sketched on top is the general two-dimensional circulation in the area, adapted from reference⁹. The WOCE transect SR4 is marked in mint, and different water masses are distinguished by colors. **b** Typical section from the Antarctic continent to the transect SR4, with general features of the overturning circulation sketched in black, adapted from references^{1,9}. Highlighted in blue are surface water mass transformations by buoyancy fluxes, and carbon fluxes are marked in green. CDW: Circumpolar Deep Water, WDW: Warm Deep Water, WSDW: Weddell Sea Deep Water, WSBW: Weddell Sea Bottom Water, DSW: Dense Shelf Water, POC: particulate organic carbon.”

Further, we have added the carbon sequestration with WSDW+WSBW to Figure 1 of the revised manuscript:

Fig. 9: Revised Fig. 1 of the manuscript.

- 4) I feel missing in the discussion a little bit more about the key role of the increased WDW over the continental shelf acting to change the seawater properties and the ability of the DSW to uptake carbon (since DSW is sourced by the intruded WDW over shelf). For example, is the proportion of DSW (30%) and WDW (70%) that accounts to form AABW (WSDW+WSBW) in the region that will predominantly change (decreasing the ability to ventilate the deep basin and, consequently, to incorporate carbon in the abyss) or the physical and biogeochemical conditions of WDW that upwelled over shelf (enriched with carbon by increased vertical carbon flux; i.e., a changing in the DSW transformation/evolution) that will prevail to reduce the sequester of carbon? A couple of paragraphs on this matter is suggested to be included.

Close to the Filchner Trough, WDW gets transported on-shore, but also dense water is exported to the abyss (see Fig. 9 above). In this region, the simulated increase in temperature and dissolved inorganic carbon (DIC) concentrations between the 1990s

and the 2090s depends on the depth and the distance to the shelf break of the water parcel in question.

While the simulated warming at 400m in *simA* amounts to $\sim 0.5^{\circ}\text{C}$ irrespective of the distance to the shelf break, the warming at 700m (corresponding to the bottom at the shelf break as defined here), is approximately four times larger close to the shelf break ($\sim 0.8^{\circ}\text{C}$) than further offshore ($0.1\text{--}0.2^{\circ}\text{C}$; see Fig. 10 a-c). Similarly, for DIC, the increase in carbon concentrations at the shelf break ($\sim 105\text{ mmol m}^{-3}$ and 90 mmol m^{-3} at 400m and 700m, respectively) exceeds the increase further offshore ($\sim 80\text{ mmol m}^{-3}$ and 50 mmol m^{-3} ; see Fig. 10 d-f).

Yet, in general, the discrepancy between the property changes at the shelf break and further offshore is larger for temperature than for DIC concentrations. This implies that physical water mass transformations on the shelf are more important for properties of waters leaving the shelf than biogeochemical transformations. The change in the carbon content over the 21st century of these waters seems to be mainly driven by the carbon content of waters transported into the southern Weddell Sea, while the warming of these waters can be largely traced to changes in the southern Weddell Sea itself.

In the revised result section, we have modified the statement about WDW, which now reads:

“Concurrently, the total heat content on the shelf increases by 66% (Fig. 6d), reflecting both an increased presence of WDW on the shelf and particularly the reduction of heat loss of shelf waters to the atmosphere, as the warming of offshore WDW is not sufficient to explain the warming of shelf waters (Supplementary Fig. 9).”

Fig. 10: Property changes of WDW in the southern Weddell Sea. **a-c** Temperature difference in $^{\circ}\text{C}$ between the 2090s and 1990s at three locations and at three depths (400 m, 700 m, and 1000 m) of the WDW core in the southern Weddell Sea as indicated by the different colors and the different hatching of the bars, respectively. See also Supplementary Fig. 16. The temperature differences are shown for *simA* (darker colors) and *simB* (lighter colors). The locations are marked in the same colors on the map in panel **c**. Black contours in the map show the 700 m (solid), 2000 m (dashed), and 3500 m (dotted) isobaths. **d-f** Same as panels **a-c**, but for the change in dissolved inorganic carbon concentrations in mmol m^{-3} . This Figure was added as Fig. 9 to the revised supplementary material.

Reviewer #3 (Remarks to the Author):

- *What are the noteworthy results?*

This paper uses a modelling framework to argue that the sequestration of carbon in the Weddell Sea can be subject to rapid changes under future climates, due to mainly physical changes associated with water mass transformations. This is important – the Weddell Sea is a key site globally where dense waters are produced which ultimately flood the global abyss, and the carbon sequestration here exerts a long-term (centennial or longer) impact on planetary-scale climate. I feel the results are noteworthy, and deserve publication – they don't "solve" the problem (and nor do the authors claim that they do), and in particular there are limitations on what can be said associated with computational expense – but they certainly add useful information and will stimulate further work (both modelling and observational) to better understand the key processes and include them in future-generation models.

- *Will the work be of significance to the field and related fields? How does it compare to the established literature? If the work is not original, please provide relevant references.*

The authors do a good job of positioning their results in the context of existing literature, and drawing out the new findings and their importance. In particular, it is worth noting that recent results by MacGilchrist et al. have enhanced attention on the open-ocean biological system in the Weddell Sea – combined with horizontal circulation, this is now seen as an important (and previously under-appreciated) vector for carbon drawdown. The present paper does not dispute/overtake this in any way, but demonstrates that predominantly physical processes happening on the shelf (and in ice shelf regions) can have profound and rapid impacts on carbon sequestration. To my mind, this stresses the importance of understanding the cross-disciplinary and cross-shelf-break system of the Weddell Sea in its totality, and incorporating that understanding into models – which the present paper does well. The authors also do a good job of reflecting model advances, from early studies that showed rapid changes in the shelf system were possible due to changing warm water intrusion, to more recent studies that reconcile this apparently more resilience. The model used in the present study aligns better with the latter studies, which is encouraging. Overall, I think the present study adds information that will be useful to climate studies, oceanography, marine productivity, and those seeking optimal routes to progress these issues – it will have broad interest.

We thank Mike Meredith for the positive feedback and for raising important points with respect to the uncertainty associated with using a single ocean-only model forced by output from a single climate model under a single emission scenario. In summary, we agree with the reviewer that some unanswered questions remain after our study regarding the integrated ocean – sea ice – ice shelf – biogeochemical system of the Weddell Sea and how each component impacts the downward transfer of carbon both in present and future climates. Hopefully, our study stimulates future modeling and observational work, in order to better constrain how this system might respond to the on-going environmental change. Please find below our detailed response to each of the points raised and a description of their inclusion in the revised manuscript.

- *Does the work support the conclusions and claims, or is additional evidence needed?*

- *Are there any flaws in the data analysis, interpretation and conclusions?*

- *Do these prohibit publication or require revision?*

- *Is the methodology sound? Does the work meet the expected standards in your field?*

I will answer these questions together...

I believe the conclusions are sound and supported, though (as with any modelling study) there are aspects that could have been explored further, had the computational resource been available. (There is always more that could have been done, right?). So I don't feel additional evidence is needed, but it might be sensible to amend the writing in places to draw

out more clearly what the current limitations are, so as to point the way forward for future studies. In particular, the issues that I feel warrant enhanced discussion are:

- Ensembles. When making projections over several decades or longer, the preferred approach is to use multiple models so that results can be drawn into ensembles and the spread examined as a means of establishing uncertainty. That isn't possible here – the model setup is unique – but the forcing for it seems to come from a single (AWI) climate model. That model is good, but what does its output look like compared to the spread of others?

We agree with the reviewer that ideally, in order to quantify the sensitivity of our results to the chosen model setup, the model experiments should be repeated using a variety of ocean models and/or forcing the same ocean model with a variety of atmospheric forcing data sets, given that the evolution of atmospheric variables over the 21st century can vary greatly depending on what climate model is used (see Fig. 11 below). Unfortunately, the calculation of such a model ensemble is beyond the scope of our study. In the absence of the computational resources to run our model experiments with output from different climate models, we have instead compiled the output of 41 climate models contributing to the 6th phase of the “Coupled Model Intercomparison Project” (CMIP6), allowing for a comparison of the evolution of globally and regionally averaged air temperatures over the 21st century in the model used in our study (AWI Climate Model) with that in other climate models (Fig. 11).

In the Weddell Sea, the multi-CMIP6-model mean suggests air temperatures of around -6.5°C in the year 2015 and a warming of just under 4°C between 2015 and 2100 (high-emission scenario SSP5-8.5; Fig. 11a & b). Yet, the spread in air temperatures is large across models. Using ± 1 standard deviation around the multi-model mean to quantify model spread, the simulated air temperatures in 2015 range from approximately -8°C to -4.5°C and the projected warming over the 21st century from 2.5°C to >5°C. While air temperatures in the AWI Climate Model are ~2°C colder in the Weddell Sea than in the multi-CMIP6-model mean, the simulated warming of just over 4°C by the end of the 21st century is fairly close to the multi-model mean. In comparison, globally, the spread across all CMIP6 models is smaller, with average air temperatures in 2015 ranging from 13.5°C to 16°C and the projected warming from 3.2°C to 5.4°C. In the AWI Climate Model, the simulated air temperatures are very close to the multi-model mean throughout the 21st century, with the projected warming in AWI Climate Model (~4°C) being only slightly lower than the multi-CMIP6-model mean (4.3°C).

Altogether, the vastly different projections of Weddell Sea air temperatures (and likely other atmospheric variables) by different climate models demonstrates that quantitatively, the projected evolution of deep-ocean carbon sequestration rates in the Weddell Sea in this study will likely be sensitive to the climate model chosen to force the ocean-only simulations. Yet, given that the chosen model agrees fairly well with the multi-model mean, the simulated mechanisms and the overall conclusions drawn from our experiments are likely robust.

In the revised manuscript, we added the Fig. 11 below (combined with Fig. 12 further down) as Fig. 20 to the revised supplementary material. Further, in the revised discussion section, we have added a statement regarding the comparison of the projected evolution of air temperatures in the AWI Climate Model as compared to other climate models:

“The AWI Climate Model used to force the experiments here projects an increase in Weddell Sea air temperatures under the SSP5-8.5 scenario (4.1°C) comparable to a multi-climate model mean (3.9°C; 2015-2100; Supplementary Fig. 20). Given the large model spread (1.3-6.6°C), using the atmospheric output of a different climate model would likely affect both the exact timing and the magnitude of the changes, but the impact on the controlling mechanisms identified here is expected to be small. Given that differences between ensemble members of the same model are typically smaller than differences between

models, using different initial conditions in our simulations would likely only affect the exact timing of the simulated changes.”

Fig. 11: Evolution of air temperatures over the 21st century under the high-emission scenario SSP5-8.5 in 41 climate models contributing to the 6th phase of the “Climate Model Intercomparison Project” (CMIP6) **a & b** in the Weddell Sea and **c & d** globally. If for any climate model, multiple ensemble members are available for the SSP5-8.5 scenario in the CMIP6 archive, only the first ensemble member is shown here. Grey lines in panels **a & c** denote individual CMIP6 models, the AWI Climate Model (as used as forcing in this study) is highlighted in blue in all panels. The black line in panels **a & c** and the grey bar in panels **b & d** denote the multi-model mean, with the standard deviation across all models being depicted as the shading and as the whisker in panels **a & c** and panels **b & d**, respectively. CMIP6 data are available via the Earth System Grid Federation (<https://esgf-data.dkrz.de/projects/cmip6-dkrz/>). This Figure was added as Fig. 20 to the revised supplementary material.

- Ensembles again. A second good approach is to re-run the model multiple times so as to test how dependant on exact starting conditions the results are. This is quite important when projecting sudden changes at end-of-century, since the timing/magnitude of those might be specific to this run, and be different in runs with subtly different starting conditions/forcings. I realise that computational power is the limiting factor here, but a statement on this limitation and what it means for interpretation of the results would be helpful.

We fully agree with the reviewer that the exact timing of any abrupt change will likely depend on the initial conditions used for the model experiments. As stated above, we are certainly more confident in the qualitative results and the mechanisms leading to the attenuated rates of carbon sequestration with Weddell Sea dense waters than in the exact timing of the simulated changes. In order to emphasize this and make the reader aware of this uncertainty, we have included it in the statement added to the revised discussion section in response to the reviewer’s preceding comment:

“The AWI Climate Model used to force the experiments here projects an increase in Weddell Sea air temperatures under the SSP5-8.5 scenario (4.1°C) comparable to a multi-climate

model mean (3.9°C; 2015-2100; Supplementary Fig. 20). Given the large model spread (1.3-6.6°C), using the atmospheric output of a different climate model would likely affect both the exact timing and the magnitude of the changes, but the impact on the controlling mechanisms identified here is expected to be small. Given that differences between ensemble members of the same model are typically smaller than differences between models, using different initial conditions in our simulations would likely only affect the exact timing of the simulated changes.”

- Scenario. The model uses a high-emissions scenario, one that actually represents higher GHG emissions than we are currently on track for. So arguably the results could represent a case that is worse than we will see, given existing (let alone future) commitments to tackle climate change. It would have been good to include an intermediate scenario and a low-emissions scenario, for comparison – though again I realise that computational power is the limiting resource. This is discussed a little bit in the text, but I feel more is warranted – in particular, if the authors agree that the likelihood of an abrupt/large-amplitude change is less under lower emissions scenarios, then a statement of that should be included.

Indeed, the SSP5-8.5 scenario assumes a continued high use of fossil fuels and weaker climate protection policies than currently in place (see IPCC). Thereby, the projected changes in, e.g., Weddell Sea air temperatures over the 21st century in the AWI Climate Model used to force the model experiments in this study are larger than in a “business-as-usual” scenario (compare to SSP2-4.5, see Fig. 12 below). In fact, the projected Weddell Sea air temperatures under the SSP5-8.5 scenario in the AWI Climate Model only diverge from those under the SSP3-7.0 and SSP2-4.5 scenarios in the late 2070s and late 2060s, respectively (Fig. 12).

However, we note that as indicated in the method section of the submitted manuscript, the atmospheric CO₂ concentrations and the resulting future climate suggested by Earth System Models under the SSP5-8.5 scenario are possible even under lower-emission scenarios as a result of the large uncertainty associated with carbon-cycle feedbacks in these models (IPCC). This uncertainty is evident from the spread of individual ensemble members for the SSP3.7-0 scenario in the AWI Climate Model, some of which hardly diverge from the SSP5-8.5 scenario at all throughout the 21st century (Fig. 12 below), and also from the large inter-model spread in Fig. 11 above (for the SSP5-8.5 scenario only). As a result, the projected changes under the SSP5-8.5 scenario in the AWI Climate Model might be the same as under more intermediate emission scenarios (such as SSP2-4.5) in another climate model. Altogether, this complicates the assessment of the likelihood of the results presented in our study, but in our view, it does not render our results impossible even if, for the remainder of this century, we are on track of a lower-emission scenario than SSP5-8.5.

In order to communicate the scenario uncertainty of our results more clearly, we have included some sentences in this regard in the revised discussion section:

“In addition, the projected evolution of atmospheric variables depends on the emission scenario used. For the AWI Climate Model, the evolution of Weddell Sea air temperatures under the SSP5-8.5 scenario diverges from that under the SSP3-7.0 scenario only after the late 2070s (Supplementary Fig. 20). While the decoupling between the continental shelf sea and the deep ocean in simA in fact accelerates around that time (Fig. 5), the earlier onset of the decoupling in the mid-21st -century is in line with the onset of declining deep-ocean carbon sequestration rates (Fig. 2 & 5). This implies that a low-emission scenario might be needed to fully avoid the decoupling and the associated attenuation in deep-ocean carbon transfer with Weddell Sea dense waters. As this conclusion is possibly model-dependent, we note that due to large uncertainties associated with carbon-cycle feedbacks in climate models, the climate resulting from a higher-emission scenario in one model is also possible under lower-emission scenarios in another⁷⁸ (see also Methods).”

Further, we have added Fig. 12 below (combined with Fig. 11 above) as Fig. 20 to the revised supplementary material.

Fig. 12: Evolution of the Weddell Sea air temperature over the 21st century in different emission scenarios of the atmospheric component of the AWI Climate Model (Semmler et al., 2020), namely SSP5-8.5 (dark blue; as used in this study), SSP3-7.0 (dark grey for the first ensemble member, light grey for all other ensemble members), SSP2-4.5 (mint), and SSP-1.26 (yellow). This Figure was added as Fig. 20 to the revised supplementary material.

- A conceptual point. The model used is a forced-ocean model, albeit it a very clever one with ice shelf cavities etc. Forcing comes from a climate model. This is all fine and well-explained, but does mean that the extra processes resolved in the fine-resolution ocean model cannot feed back onto the large-scale climate that forces it. I doubt this is a particular factor – the likely climate feedbacks will be centennial or longer scales, not decadal scales that the model is run for – but I think it is worth including a statement on this in the paper.

We thank the reviewer for raising this point. Indeed, the ocean-only model employed here disregards any feedbacks of changes in oceanic carbon cycling with the atmosphere and hence climate, which would possibly become increasingly important on time scales longer than those considered here. In the revised manuscript, we have added a statement along those lines in the discussion section, which now reads:

“On centennial and millennial time scales, a reduction in deep-ocean carbon accumulation will induce climate feedbacks as a consequence of the reduced storage time-scale. While these likely matter only marginally on the time scales considered in this study, justifying the use of an ocean-only model setup, longer simulations with fully coupled climate models including ice-shelf cavities should quantify both the magnitude of and the time scales associated with these feedbacks in future work. “

- Is there enough detail provided in the methods for the work to be reproduced?

Yes, but in reality nobody will try to reproduce exactly these results given the complexity of the model and computational resource needed to run it. But I feel that it will stimulate further studies (including by other groups) who will seek to reproduce the basics of this study as part of developing the science further and more fully.

A couple of extra points:

- The paper highlights the extra presence of WDW on the shelf as a causal factor in the abrupt changes, but does not go into much detail concerning the mechanisms that enable this. (There is some, but not much). I presume these are wind forced changes, reduced sea ice production (and hence shallower mixed layer, less dense shelf water) etc. A little more detail here would be helpful to the reader.

In response to this comment, we refer the reviewer to a comment along similar lines by reviewer #2 (see his/her comment #4 above). In particular, we have added Fig. 10 above to the revised Supplementary material, demonstrating that for temperature changes between the 1990s and the 2090s, changes in water mass transformations on the shelf dominate over property changes in WDW offshore.

We have not yet gone into more detail in our analysis regarding the relative role of changes in the wind forcing, changes in sea ice production, and mixed layer depth in controlling these property changes, but we agree with the reviewer that this is an exciting aspect of future research.

- By comparison, a whole paragraph in the discussion is given over to comparing the performance of the model re AABW export with that in ESMs. I think this can be condensed? (Nobody sensible believes that coarse-resolution ESMs do a good job of AABW formation/export, do they?)

We have revisited the paragraph in question for the revised manuscript and have shortened the discussion on the comparison of AABW formation in different models. The first paragraph of the discussion section now reads:

“Using a model setup that, for the first time, includes both ice-shelf cavities and a representation of the ocean carbon cycle, our results quantify the role of both physical and biological processes in deep-ocean carbon accumulation in the southern Weddell Sea. Over most of the 21st century, the carbon transfer to the abyss due to physical processes is up to four times higher than due to biological processes in the high-emission scenario, until the physically-driven transfer abruptly changes sign in the 2090s (Fig. 2). This is the result of an increasing decoupling of bottom waters on the continental shelves and the deep ocean since the mid-century (Fig. 2 & 5), with bottom water density declining most along pathways of dense water export from the southern continental shelves to the abyss (Filchner Trough; see Fig. 1 & 6). Ultimately, Weddell Sea Deep and Bottom Waters are thereby effectively cut off from renewal with waters descending the continental slope, as the newly formed lighter dense waters end up at shallower depths. Notably, the decoupling of bottom waters on the southern continental shelves and the open ocean is sustained in two idealized model extensions, which were run for 50 years and forced with constant end-of-century air temperatures (ext1 and ext2; see Supplementary Section 1, Supplementary Fig. 13-16, and Methods). While episodic open-ocean mixing events temporarily enhance deep-ocean carbon accumulation in our experiments, the scarcity of adequate observations prevents the evaluation of the present-day frequency of these events and their impact on the deep-ocean carbon budget of the southern Weddell Sea. Given their absence in simC (Fig. 2d), these open-ocean mixing events seem to mainly be triggered by climate variability, but whether the frequency or duration of these events changes over the 21st century remains unresolved (due to the lack of physical flux output for some decades; see Methods). Overall, the simulated decline in dense-water transfer to the deep ocean in response to enhanced stratification is in line with previous modelling experiments⁷. However, in contrast to some Earth System Models, which often exclusively form bottom waters via open-ocean mixing in areas north of the transect considered in this study⁷, our model does not suggest a complete shutdown of dense water formation by 2100, but a less efficient transfer of newly formed dense waters from the Weddell Sea continental shelves to the abyss (Fig. 8). Although the downstream effects of the two mechanisms may be similar, namely less carbon sequestration in the high-latitude Southern Ocean, bottom water renewal is known to predominantly occur along the continental margins¹³⁻¹⁶. Thus, our model experiments display a more realistic representation of the mechanisms involved in dense water formation and transfer to the deep ocean, which

is indispensable when aiming to anticipate their response to the on-going environmental change.”

- “Cumulative accumulation” sounds odd to me, for some reason. Why not just “Accumulated....”?

Changed as suggested throughout the manuscript (see also Fig. 2 above).

Cited references

- Akaike, H. (1974). A new look at the statistical model identification. *IEEE Transactions on Automatic Control*, 19(6), 716–723. <https://doi.org/10.1109/TAC.1974.1100705>
- Beaulieu, C., & Killick, R. (2018). Distinguishing Trends and Shifts from Memory in Climate Data. *Journal of Climate*, 31(23), 9519–9543. <https://doi.org/10.1175/JCLI-D-17-0863.1>
- Karakuş, O., Völker, C., Iversen, M., Hagen, W., Wolf-Gladrow, D., Fach, B., & Hauck, J. (2021). Modeling the Impact of Macrozooplankton on Carbon Export Production in the Southern Ocean. *Journal of Geophysical Research: Oceans*, 126(12), 1–22. <https://doi.org/10.1029/2021JC017315>
- Killick, R., Beaulieu, C., Taylor, S., and Hullait, H. (2021). EnvCpt: Detection of structural changes in climate and environment time series. R package version 1.1.3, <https://cran.r-project.org/web/packages/EnvCpt/EnvCpt.pdf>
- Marshall, J., & Speer, K. (2012). Closure of the meridional overturning circulation through Southern Ocean upwelling. *Nature Geoscience*, 5(3), 171–180. <https://doi.org/10.1038/ngeo1391>
- Semmler, T., Danilov, S., Gierz, P., Goessling, H. F., Hegewald, J., Hinrichs, C., Koldunov, N., Khosravi, N., Mu, L., Rackow, T., Sein, D. V., Sidorenko, D., Wang, Q., & Jung, T. (2020). Simulations for CMIP6 With the AWI Climate Model AWI-CM-1-1. *Journal of Advances in Modeling Earth Systems*, 12(9), 1–34. <https://doi.org/10.1029/2019MS002009>
- Silvano, A., Rintoul, S. R., Peña-Molino, B., Hobbs, W. R., van Wijk, E., Aoki, S., Tamura, T., & Williams, G. D. (2018). Freshening by glacial meltwater enhances melting of ice shelves and reduces formation of Antarctic Bottom Water. *Science Advances*, 4(4), 1–12. <https://doi.org/10.1126/sciadv.aap9467>
- Thomas, Z. A., Jones, R. T., Turney, C. S. M., Golledge, N., Fogwill, C., Bradshaw, C. J. A., Menviel, L., McKay, N. P., Bird, M., Palmer, J., Kershaw, P., Wilmshurst, J., & Muscheler, R. (2020). Tipping elements and amplified polar warming during the Last Interglacial. *Quaternary Science Reviews*, 233. <https://doi.org/10.1016/j.quascirev.2020.106222>
- Turney, C. S. M., Fogwill, C. J., Golledge, N. R., McKay, N. P., van Sebille, E., Jones, R. T., Etheridge, D., Rubino, M., Thornton, D. P., Davies, S. M., Ramsey, C. B., Thomas, Z. A., Bird, M. I., Munksgaard, N. C., Kohno, M., Woodward, J., Winter, K., Weyrich, L. S., Rootes, C. M., ... Cooper, A. (2020). Early Last Interglacial ocean warming drove substantial ice mass loss from Antarctica. *Proceedings of the National Academy of Sciences of the United States of America*, 117(8), 3996–4006. <https://doi.org/10.1073/pnas.1902469117>
- Vernet, M., Geibert, W., Hoppema, M., Brown, P. J., Haas, C., Hellmer, H. H., Jokat, W., Jullion, L., Mazloff, M., Bakker, D. C. E., Brearley, J. A., Croot, P., Hattermann, T., Hauck, J., Hillenbrand, C. -D., Hoppe, C. J. M., Huhn, O., Koch, B. P., Lechtenfeld, O. J., ... Verdy, A. (2019). The Weddell Gyre, Southern Ocean: Present Knowledge and Future Challenges. *Reviews of Geophysics*, 57(3), 623–708. <https://doi.org/10.1029/2018RG000604>

REVIEWERS' COMMENTS

Reviewer #1 (Remarks to the Author):

Second review of “Abruptly attenuated carbon sequestration with Weddell Sea dense waters towards the end of the 21st century” by Nissen et al.

Firstly, my apologies for the lateness of this review - this turned out to be a busier month than anticipated.

The authors have provided a comprehensive and robust rebuttal to my primary concern - namely that the analysis was inconclusive in addressing whether conditions at the end of the 21st century are notably distinct from decadal variability and would be sustained. I believe that the change point analysis, the extended simulations, and the closer inspection of changing shelf-to-open ocean properties, all confirm that the transfer of newly ventilated water to the deep ocean (and associated carbon sequestration) has been significantly reduced by the end of the century. Moreover, it seems unlikely that sequestration will not recover on a decadal timescale. I concur with the proposed mechanism by which this has taken place - namely the lightening of dense shelf waters (primarily by freshening) that no longer sink into the abyss.

My ongoing concerns relate to the continued comparison to the 2080s and the assertion that the change in the 2090s is “abrupt”. It appears that, in the latter half of the 21st century, an instability emerges that leads to episodic open ocean mixing events - a common occurrence for models of this region*. This is confirmed by the extension experiments in which such events continue to occur (particularly in ext2). These events clearly have the potential to strongly impact carbon sequestration in the deep ocean, and one such event is the primary driver for the elevated accumulation in the 2080s.

I would suggest that it is only by comparison to this anomalous period that the decline in the 2090s could be considered “abrupt”. Instead, what occurs in the 2090s appears to be part of an ongoing gradual decline that has been taking place since the 1950s — a fact that the authors acknowledge, and which is supported by the change point analysis. The onset of the episodic open ocean mixing event appears to obscure the more gradual forced dynamics (declining bottom water formation and carbon sequestration) taking place in the region.

I don't think this is a significant roadblock for the publication of the paper. The simulation, analysis and discussion of the changes are well-presented and valuable contributions that will inspire further work. Furthermore, I don't think it detracts from the interest or impact of the paper to note that the change is gradual rather than abrupt. However, I would suggest that the authors should consider whether this aspect of the paper's framing — the abruptness of the attenuation — is a fair reflection of what is occurring in the model. I would suggest removing direct comparisons to the 2080s (e.g. in the abstract, line 93, line 143, Figure 3), and references to the abruptness of what happens in the 2090s, including in the title of the paper. Furthermore, a more central discussion of the 2080s event as an anomaly overlying a background trend would give a fuller perspective to the reader. This is mentioned in parts of the manuscript, but usually as an aside (e.g. lines 282 to 285), while the notion of the 2090s as the moment of abrupt change is retained.

I don't wish to exert outsized influence on the paper's framing. I am absolutely open to the authors arguing that the apparent "abruptness" of the 2090s change arises not simply by comparison to an anomalous 2080s, but is a distinct state change in itself. I don't yet see evidence for that, and the additional analysis - while confirming clear overall and sustained changes, as well as the authors' proposed mechanisms - appears instead to affirm that the forced change over the latter half of the 21st century is gradual and that what happens in the 2080s is the anomaly.

Sincerely,
Graeme MacGilchrist

*As an aside, the authors may wish to also cite work by Lockwood et al. (2021) *Journal of Climate*, 34(7), who note that, when boundary currents are resolved and freshwater anomalies constrained to the shelf (as in this high-resolution simulation), open ocean deep mixing may continue to occur, in contrast to the findings of de Lavergne et al.

Reviewer #2 (Remarks to the Author):

Second review of the manuscript "Abruptly attenuated carbon sequestration with Weddell Sea dense waters towards the end of the 21st century" by Nissen et al. submitted to *Nature Communications*.

I am satisfied with the review conducted by the authors, who satisfactorily answered the doubts raised and clarified the message in the text, such as requested. Thus, I recommend acceptance of the manuscript for publication as presented.

Reviewer #3 (Remarks to the Author):

I feel the authors have done a good job of responding to my comments on their initial submission - they have strengthened the manuscript in the appropriate places, and provided detailed responses outlining their reasoning behind the changes made. I don't have further concerns, and, from my perspective, I feel the paper is ready for publication. There are various small issues that could be debated, but the argument the authors make is quite clear and coherent, and others can build upon/challenge the work in future papers should they choose.

Response to comments by reviewers

Reviewer #1 (Remarks to the Author):

Second review of “Abruptly attenuated carbon sequestration with Weddell Sea dense waters towards the end of the 21st century” by Nissen et al.

Firstly, my apologies for the lateness of this review - this turned out to be a busier month than anticipated.

The authors have provided a comprehensive and robust rebuttal to my primary concern - namely that the analysis was inconclusive in addressing whether conditions at the end of the 21st century are notably distinct from decadal variability and would be sustained. I believe that the change point analysis, the extended simulations, and the closer inspection of changing shelf-to-open ocean properties, all confirm that the transfer of newly ventilated water to the deep ocean (and associated carbon sequestration) has been significantly reduced by the end of the century. Moreover, it seems unlikely that sequestration will not recover on a decadal timescale. I concur with the proposed mechanism by which this has taken place - namely the lightening of dense shelf waters (primarily by freshening) that no longer sink into the abyss.

My ongoing concerns relate to the continued comparison to the 2080s and the assertion that the change in the 2090s is “abrupt”. It appears that, in the latter half of the 21st century, an instability emerges that leads to episodic open ocean mixing events - a common occurrence for models of this region*. This is confirmed by the extension experiments in which such events continue to occur (particularly in ext2). These events clearly have the potential to strongly impact carbon sequestration in the deep ocean, and one such event is the primary driver for the elevated accumulation in the 2080s.

I would suggest that it is only by comparison to this anomalous period that the decline in the 2090s could be considered “abrupt”. Instead, what occurs in the 2090s appears to be part of an ongoing gradual decline that has been taking place since the 1950s — a fact that the authors acknowledge, and which is supported by the change point analysis. The onset of the episodic open ocean mixing event appears to obscure the more gradual forced dynamics (declining bottom water formation and carbon sequestration) taking place in the region.

I don't think this is a significant roadblock for the publication of the paper. The simulation, analysis and discussion of the changes are well-presented and valuable contributions that will inspire further work. Furthermore, I don't think it detracts from the interest or impact of the paper to note that the change is gradual rather than abrupt. However, I would suggest that the authors should consider whether this aspect of the paper's framing — the abruptness of the attenuation — is a fair reflection of what is occurring in the model. I would suggest removing direct comparisons to the 2080s (e.g. in the abstract, line 93, line 143, Figure 3), and references to the abruptness of what happens in the 2090s, including in the title of the paper. Furthermore, a more central discussion of the 2080s event as an anomaly overlying a background trend would give a fuller perspective to the reader. This is mentioned in parts of the manuscript, but usually as an aside (e.g. lines 282 to 285), while the notion of the 2090s as the moment of abrupt change is retained.

I don't wish to exert outsized influence on the paper's framing. I am absolutely open to the authors arguing that the apparent “abruptness” of the 2090s change arises not simply by comparison to an anomalous 2080s, but is a distinct state change in itself. I don't yet see evidence for that, and the additional analysis - while confirming clear overall and sustained changes, as well as the authors' proposed mechanisms - appears instead to affirm that the forced change over the latter half of the 21st century is gradual and that what happens in the 2080s is the anomaly.

Sincerely,
Graeme MacGilchrist

*As an aside, the authors may wish to also cite work by Lockwood et al. (2021) Journal of Climate, 34(7), who note that, when boundary currents are resolved and freshwater anomalies constrained to the shelf (as in this high-resolution simulation), open ocean deep mixing may continue to occur, in contrast to the findings of de Lavergne et al.

We thank Graeme MacGilchrist for his comments on the revised version of our manuscript.

In general, we agree with the points raised by him, but still think that our interpretation of the simulated changes is valid. Nonetheless, in response to the reviewer's comment, we have added a paragraph to the discussion section to more clearly highlight this alternative view on our data. It reads:

“The view that the attenuation of carbon sequestration rates in the 2090s is abrupt implies a system change in the years directly leading up to this decade. Here, we interpret the fact that all physical fluxes combined act to transfer carbon upwards instead of into the deep ocean for the first time in this final decade of the 21st century as evidence for such a recent system change. In particular, compared to the physical fluxes throughout the 21st century, their change in the 2090s is drastic (Fig. 2g). As a result, the deep-ocean carbon accumulation in the 2090s is below the average rate over the whole analysis period ± 1 standard deviation for the first time in this final decade (Fig. 2e). Opposing to this view, the reduced sequestration rate in the 2090s can admittedly also be interpreted as a result of a gradual system change over a longer time period, with the high sequestration rates in the 2080s being a temporary anomaly (supported by the change point analysis; see Fig. 2). Nonetheless, whether gradual or abrupt, the system is clearly in a different state at the end of the 21st century.”

Furthermore, to put less emphasis on the comparison of the 2090s with the 2080s alone, we have added a reference to the 2050s in both the abstract and the result section. In our view, this emphasizes that the conditions in the 2090s are not only different when comparing to the admittedly anomalous 2080s, but also to preceding decades. The respective sentences read as follows:

Abstract:

„Here, we use a model setup including both ice-shelf cavities and oceanic carbon cycling and demonstrate that by 2100, deep-ocean carbon accumulation in the southern Weddell Sea is abruptly attenuated to only 40% of the 1990s rate in a high-emission scenario, while the rate in the 2050s and 2080s is still 2.5-fold and 4-fold higher, respectively, than in the 1990s.“

Results:

“Acknowledging substantial decadal and interannual variability (Fig. 2c), the average deep-ocean accumulation rate of carbon amounts to 3.7 Tg C yr⁻¹ in the 1990s, increases to 9.4 Tg C yr⁻¹ and 14.7 Tg C yr⁻¹ in the 2050s and 2080s, respectively, and then abruptly declines to 1.5 Tg C yr⁻¹ in the 2090s (Fig. 2e).”

Lastly, we thank the reviewer for suggesting to include the very relevant study by Lockwood et al. (2021) in our reference list. We have included it in the revised discussion section, and the respective part now reads:

“Given their absence in simC (Fig. 2d), these open-ocean mixing events seem to mainly be triggered by climate variability. While it remains unresolved whether the frequency or duration of these events changes over the 21st century (due to the lack of physical flux output for some decades; see Methods), we note that the continued occurrence of these events in a warmer climate is in agreement with a recent study using another high-resolution model³⁹.”

Reviewer #2 (Remarks to the Author):

Second review of the manuscript "Abruptly attenuated carbon sequestration with Weddell Sea dense waters towards the end of the 21st century" by Nissen et al. submitted to Nature Communications.

I am satisfied with the review conducted by the authors, who satisfactorily answered the doubts raised and clarified the message in the text, such as requested. Thus, I recommend acceptance of the manuscript for publication as presented.

We are happy to hear that reviewer #2 is satisfied with the changes we applied to the manuscript in response to his/her comments.

Reviewer #3 (Remarks to the Author):

I feel the authors have done a good job of responding to my comments on their initial submission - they have strengthened the manuscript in the appropriate places, and provided detailed responses outlining their reasoning behind the changes made. I don't have further concerns, and, from my perspective, I feel the paper is ready for publication. There are various small issues that could be debated, but the argument the authors make is quite clear and coherent, and others can build upon/challenge the work in future papers should they choose.

We are happy to hear that reviewer #3 is satisfied with the changes we applied to the manuscript in response to his/her comments.